# PLAN2EVOLVE: LLM SELF-EVOLUTION FOR IMPROVED PLANNING CAPABILITY VIA AUTOMATED DOMAIN GENERATION

## ABSTRACT

Large Language Models (LLMs) have recently shown strong potential in robotic task planning, particularly through automatic planning domain generation that integrates symbolic search. Prior approaches, however, have largely treated these domains as search utilities, with limited attention to their potential as scalable sources of reasoning data. At the same time, progress in reasoning LLMs has been driven by chain-of-thought (CoT) supervision, whose application in robotics remains dependent on costly, human-curated datasets. We propose Plan2Evolve, an LLM self-evolving framework in which the base model generates planning domains that serve as engines for producing symbolic problem–plan pairs as reasoning traces. These pairs are then transformed into extended CoT trajectories by the same model through natural-language explanations, thereby explicitly aligning symbolic planning structures with natural language reasoning. The resulting data extend beyond the model's intrinsic planning capacity, enabling model fine-tuning that yields a planning-enhanced LLM with improved planning success, stronger cross-task generalization, and reduced inference costs.

## 1 INTRODUCTION

Large Language Models (LLMs) have shown strong potential in robotic task planning due to their reasoning capabilities and cross-task generalization (Huang et al., 2023; 2022; Wang et al., 2024; Li et al., 2023; Zhao et al., 2024). However, like other learning-based methods, LLMs often suffer from stochastic outputs and error accumulation over long-horizon tasks, leading to failure in verifying action feasibility, state tracking, and dependency verification. To resolve these issues, researchers have combined LLMs with symbolic search-based algorithms for better long-horizon planning performance (Meng et al., 2024; Hu et al., 2023; Liu et al., 2023a). More recently, LLMs have been used to automatically infer planning domains, which, once combined with tree search, yield task-specific planners (Oswald et al., 2025; Byrnes et al., 2024; Guan et al., 2023b; Han et al., 2024a; Huang et al., 2025b). While effective, the domain-inference approaches mainly treat planning domains as mere search utilities, overlooking their potential to serve as generative engines of scalable, verifiable training data (Dalal et al., 2023; Khodeir et al., 2023).

In parallel, reasoning-focused LLMs have advanced significantly through chain-of-thought (CoT) training, where models learn to decompose complex problems into multi-step reasoning traces (Wei et al., 2022; Cobbe et al., 2021; Zelikman et al., 2022), yet CoT training usually relies on large, manually curated datasets, which are costly to construct, especially for robotics. A promising alternative is to leverage symbolic structures for scalable data production (Dalal et al., 2023). However, while prior research has extensively studied language-to-symbol transformation for robotic planning (Pan et al., 2023b; Han et al., 2024b; Tafjord et al., 2021), the reverse process, symbol-to-language transformation, remains underexplored for model trainingWang et al. (2025a). This gap raises an important question: can LLMs bootstrap their own planning ability by aligning symbolic plans with natural-language reasoning to internalize planning ability?

We propose PLAN2EVOLVE, a self-evolving framework that reinterprets LLM-generated planning domains as knowledge sources for data generation. Given diverse tasks, the base model first self-generates planning domains that produce problem–plan pairs. These symbolic plans are then explained

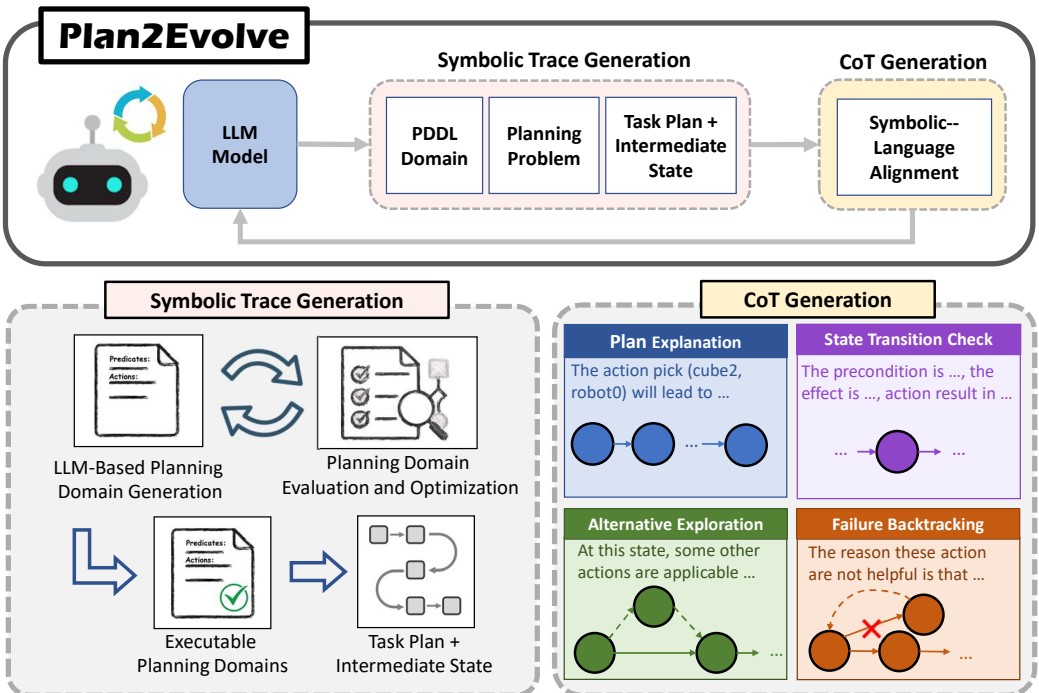

Figure 1: Overview of the proposed PLAN2EVOLVE framework. The base model first induces symbolic planning domains via LLM-based PDDL generation and optimization, which serve as data generators to produce solution plans together with their intermediate state transitions, forming problem–plan pairs. These pairs are then transformed by the same base model into CoT reasoning traces through plan explanation, state transition checking, alternative exploration, and failure backtracking, thereby aligning symbolic structures with the base-model internal natural-language reasoning. Fine-tuning on this self-generated corpus internalizes symbolic understanding and planning behavior, which yields a planning-enhanced LLM with stronger generalization and planning capability.

by the same model in natural language, resulting in long CoT trajectories aligned with formal planning semantics. Together, these outputs form a unified dataset that extends beyond the intrinsic planning capability of the base model, enabling supervised fine-tuning (SFT) to internalize and enhance its planning performance. Our contributions are:

- **PLAN2EVOLVE Framework:** We introduce PLAN2EVOLVE, a novel framework that treats LLM-generated PDDL domains as evolving knowledge sources, whose compositionality enables systematic task generation and the automatic creation of scalable planning supervision.

- **Self-evolving Data Generation:** PLAN2EVOLVE enables the base model to generate validated long-horizon robotic planning problem–plan pairs that go beyond its intrinsic planning capacity, thereby eliminating the need for human curation of training data.

- **Symbolic–Language Alignment:** We introduce an automatic self-alignment procedure that translates symbolic PDDL plans and states into natural-language CoT using the base model, and empirically demonstrate that symbolic–language alignment is essential for training LLM to plan, with higher-quality alignment yielding substantial performance gains.

- **Empirical Gains:** By SFT, PLAN2EVOLVE produces a planning-enhanced LLM that achieves robust planning performance, stronger cross-task generalization, and reduced inference token costs. These results indicate that PLAN2EVOLVE can improve both the efficiency and planning capacity of LLM.

## 2 RELATED WORK

**Learning to plan** LLMs have emerged as powerful tools for robotic task planning (Huang et al., 2023; 2022; Wang et al., 2024; Chen et al., 2024; Li et al., 2023; Zhao et al., 2024). Early efforts treated LLMs as direct planners, but models struggle with long-horizon dependencies (Wang et al., 2024; Sermanet et al., 2023; Driess et al., 2023; Brohan et al., 2023; Chen et al., 2023; Wang et al., 2024). Subsequent work has used LLMs to guide symbolic search, improving exploration efficiency while maintaining completeness (Zhao et al., 2024; Yang et al., 2024; Meng et al., 2024; Hu et al., 2023; Silver et al., 2024), yet they rely on manually engineered planning domain or search trees, limiting scalability. A complementary direction investigates automatic planning domain generation in Planning Domain Definition Language (PDDL) (McDermott et al., 1998) , where symbolic representations of the world are learned from data or inferred by LLMs through interactions. Some approaches refine partial domains (Diehl et al., 2021; Kumar et al., 2023; Silver et al., 2023; Liang et al., 2024; Athalye et al., 2024; Byrnes et al., 2024; Wong et al., 2023; Liu et al., 2025; Zhu et al., 2024; Huang et al., 2025a), while others construct domains from natural-language descriptions (Guan et al., 2023b; Han et al., 2024a; Oswald et al., 2025). More recently, Huang et al. (2025b) demonstrated that demonstration trajectory can directly induce planning domains. Although these advances prove LLMs as capable domain generators, they regard PDDL domains as search utilities rather than as scalable engines for producing verified reasoning data. Prior work shows that PDDL can guide robot motion generation for imitation learning (Dalal et al., 2023; Khodeir et al., 2023), highlighting its potential as an LLM training source, too. Thus, we develop a self-evolving framework that leverages its own generated planning domains as a source of training data, and demonstrate its strong effectiveness.

**Post-training to LLMs with Chain of Thought** Early progress in reasoning with LLMs relied on few-shot prompting but failed on complex multi-step problems (Brown et al., 2020). CoT prompting (Wei et al., 2022) demonstrated that explicit step-by-step reasoning traces improve performance (Kojima et al., 2022). Later, inference-time strategies were developed to further strengthen reasoning (Wang et al., 2023a; Yao et al., 2023). Beyond prompting, SFT (Ouyang et al., 2022) has been used to train models to generate and validate CoT reasoning traces (Cobbe et al., 2021), while correctness-validated rationales have yielded significant improvements in reasoning performance (Zelikman et al., 2022; Yuan et al., 2023; Tong et al., 2024; Lee et al., 2025; Hosseini et al., 2024; Wang et al., 2025b). Despite recent advances, LLMs still struggle to interpret symbolic representations (Patel et al., 2024; Liu et al., 2023b). Most prior work has focused on language-symbol transformations (Pan et al., 2023b; Han et al., 2024b; Olausson et al., 2023; Xu et al., 2024; Pan et al., 2023a; Liu et al., 2023a; Tafjord et al., 2021), whereas efforts on symbol-language transformation remain limited (Wang et al., 2025a; Feng et al., 2024), particularly in robotics. Recent studies reveal that symbolic planning often aligns poorly with natural-language traces (Stechly et al., 2024), creating a barrier to leveraging symbolic plan traces as training data to improve LLM planning. To bridge this gap, we propose an automatic symbolic–language alignment method via LLM self-alignment strategy. We show that the quality of this alignment is critical for fine-tuning and significantly influences fine-tuned LLM planning performance.

## 3 PRELIMINARIES

### 3.1 SYMBOLIC FORMALIZATION OF PDDL (PLANNING DOMAIN DEFINITION LANGUAGE)

A PDDL domain is defined by $\mathcal{D} = (\mathcal{P}, \mathcal{A})$, where $\mathcal{P}$ is a set of predicate and $\mathcal{A}$ is a set of action. The problem-specific object set is defined as $\mathcal{O} = \{o_1, \ldots, o_n\}$. Each **predicate** $p \in \mathcal{P}$ describes object property or relations characterized by a Boolean classifier $p(o_1, \ldots, o_i) \rightarrow \{0, 1\}$. Instantiating $p$ with concrete objects $o_1, \ldots, o_i \in \mathcal{O}$ yields a ground atom. Let

$$\mathcal{G} = \{ p(o_1, \ldots, o_k) \mid p \in \mathcal{P}, \ o_i \in \mathcal{O} \} \tag{1}$$

be the set of all possible ground atoms. A symbolic state is a set of true atoms, $\mathcal{X} \subseteq \mathcal{G}$. An **action** $a \in \mathcal{A}$ is defined as $a = \langle \text{PRE}, \text{EFF}^+, \text{EFF}^- \rangle$, where PRE denotes the predicates that must hold for the action to be applicable. The effects consist of add effects $\text{EFF}^+$ and delete effects $\text{EFF}^-$, which specify how the state is updated when the action is executed. Initiating $a$ with concrete objects

$o_1, \ldots, o_j \in \mathcal{O}$ results in a ground action $a(o_1, \ldots, o_j)$. Executing $a$ induces the state transition $\mathcal{X}^t \times a \to \mathcal{X}^{t+1}$

**Planning problem.** Thus, a formal definition of a planning problem becomes,

$$\mathcal{Q} = \langle \mathcal{O}, \mathcal{D}, \mathcal{X}^{(init)}, \mathcal{X}^{(goal)} \rangle, \mathcal{D} = (\mathcal{P}, \mathcal{A}), \tag{2}$$

and a solution plan is a sequence of ground actions

$$\tau = \{a^{(0)}, \ldots, a^{(T-1)}\} = \texttt{PDDLSolver}(\mathcal{Q}), \forall a^{(i)} \in \mathcal{A} \tag{3}$$

such that $\mathcal{X}^{(init)} \times \tau \to \mathcal{X}^{(goal)}$.

### 3.2 AUTOMATIC LLM PLANNING DOMAIN GENERATION

A base LLM $\mathcal{M}_0$ can automatically generate planning domains from unstructured inputs such as natural language descriptions, task specifications, or demonstrations (depending on the exact method adopted). Formally,

$$(\hat{\mathcal{P}}, \hat{\mathcal{A}}) = \Psi^{\mathcal{M}_0}(\mathcal{U}), \tag{4}$$

where $\mathcal{U}$ denotes the input source and $\hat{\mathcal{D}}$ is the generated domain with predicates $\hat{\mathcal{P}}$ and actions $\hat{\mathcal{A}}$. $\Psi$ indicates the selected domain generation method. The resulting domain enables planning problems to be solved by symbolic search.

### 3.3 FROM PDDL TO CoT GENERATION

Given a generated domain $\hat{\mathcal{D}}$, the symbolic search naturally induces structured reasoning traces. A solution plan $\tau = \{a^{(0)}, \ldots, a^{(T-1)}\}$ encodes a verifiable sequence of state transitions that can be expanded into stepwise natural-language explanations.

**Symbolic trace.** For a planning problem $\mathcal{Q} = \langle \mathcal{O}, \hat{\mathcal{D}}, \mathcal{X}^{(init)}, \mathcal{X}^{(goal)} \rangle$ and its solution $\tau$, each ground action $a^{(t)}$ yields the ordered symbolic state transition trace

$$\mathcal{T}^{sym} = \{(\mathcal{X}^t, a^{(t)}, \mathcal{X}^{t+1})\}_{t=0}^{T-1}. \tag{5}$$

**Natural-language trace.** Each symbolic state transition trace $(\mathcal{X}^t, a^{(t)}, \mathcal{X}^{t+1})$ is then mapped by the base model $\mathcal{M}_0$ into a natural-language explanation:

$$e^{(t)} = f_{\text{NL}}^{\mathcal{M}_0}\left(\mathcal{X}^t, a^{(t)}, \mathcal{X}^{t+1}\right). \tag{6}$$

**CoT trajectory.** Concatenating these explanations yields the full CoT trajectory

$$\text{CoT}_\tau = \{e^{(0)}, e^{(1)}, \ldots, e^{(T-1)}\}, \tag{7}$$

which explicitly aligns symbolic planning semantics with natural-language reasoning.

## 4 PROBLEM SETTING

We address the problem of evolving a base language model $\mathcal{M}_0$ into a planning-enhanced model $\mathcal{M}_{P2E}$ through the PLAN2EVOLVE framework. Given domain inference input $\mathcal{U}$, the base model $\mathcal{M}_0$ induces a symbolic planning domain $\hat{\mathcal{D}}$, on which a symbolic solver generates problem–plan pairs $\langle \mathcal{Q}, \tau \rangle$. Each plan $\tau$ is decomposed into state transitions that $\mathcal{M}_0$ maps into CoT explanations $\text{CoT}_\tau$. The paired data are then concatenated into full reasoning trace $\zeta_{align} = \langle \mathcal{Q}, \tau, \text{CoT}_\tau \rangle$ which serve as training sources. Aggregated over tasks, the collection $\mathcal{C} = \{\zeta_{align}^i\}_{i=1}^N$ forms a corpus for model fine-tuning, yielding the evolved model $\mathcal{M}_{P2E}$ with improved planning ability, stronger generalization, and enhanced long-horizon reasoning.

## 5 METHODOLOGY

PLAN2EVOLVE is a self-evolving framework that leverages a base LLM $\mathcal{M}_0$ to generate and refine PDDL domains, which are then used with a symbolic planner to produce problem–plan pairs. The base LLM then performs symbolic–language alignment by explaining these pairs as CoT reasoning traces. Through SFT on these aligned traces, $\mathcal{M}_0$ internalizes the search process and strengthens its planning capability, generalization, and efficiency. Details are explained below.

## 5.1 AUTOMATIC PLANNING DOMAIN GENERATION

The first stage of PLAN2EVOLVE is the automatic generation of symbolic planning domains. Leveraging the generative capacity and reasoning ability of $\mathcal{M}_0$, the LLM proposes predicates and actions that are consistent with the input $\mathcal{U}$. To generate a consistent and executable PDDL domain, we adopt a two-step generation procedure. First, we employ an LLM-based domain generation method by Huang et al. (2025b) to construct an initial domain skeleton with $\mathcal{M}_0$, which is then validated against planning problems. Second, when validation fails due to domain error, we introduce two complementary self-correction mechanisms. On one hand, error traces from the symbolic solver are reformulated as diagnostic feedback to prompts $\mathcal{M}_0$, enabling mistake rectification and domain repair (Guan et al., 2023a; Oswald et al., 2025). On the other hand, a hill-climbing algorithm (Silver et al., 2023; Kumar et al., 2023) is employed to prune redundant components from the domain. Details of these algorithms can be found in appendix. Once PDDL planning domains are generated, they can automatically solve planning problems to generate the symbolic problem-plan pairs. Full details provided in Section B.1.

## 5.2 SYMBOLIC-LANGUAGE ALIGNMENT

The next step is to establish an alignment between symbolic plans and natural language reasoning. This step is crucial: directly training the LLM to raw symbolic structures often leads to minimal performance improvement or even degradation, as we later demonstrate experimentally. Given a planning problem $\mathcal{Q}$ and its solution plan $\tau$, the base model $\mathcal{M}_0$ is prompted with a structured template to provide natural-language explanations of the symbolic transitions, thereby constructing symbolic–language alignment. This alignment is not restricted to a linear narration of executed actions. Instead, it forces the LLM to capture stepwise symbolic search behaviors in natural language across four complementary aspects. **Plan explanation** justifies the chosen sequence of actions on how each action contributes to overall goal. **State transition check** enforces verifying if action preconditions are satisfied and apply action effects. **Alternative exploration** considers other applicable actions in the current state and simulate their resulting state. **Failure backtracking** diagnoses why certain actions are infeasible or fail to contribute to the overall goal, and reverting to earlier states to explore alternative choices. By translating symbolic structures into natural language CoTs, the model grounds symbolic syntax in its latent reasoning space, thereby enabling enhancement of planning capabilities by fine-tuning. Details on prompt template can be found at Section B.3.2.

## 5.3 DATA CONSTRUCTION

The training corpus is automatically constructed by integrating symbolic problem–plan pairs with their corresponding symbolic-language aligned CoT. To enhance robustness and reduce overfitting to a single solution trajectory, the PDDL solver is allowed to generate diverse solution paths for the same problem, including both optimal plans and suboptimal variants containing redundant or corrective steps. Such augmentation exposes the model to a broader range of planning strategies, capturing trade-offs between plan optimality and reasoning complexity, and error correction behaviors. The resulting dataset thus encapsulates a wide spectrum of reasoning patterns, forming rich supervision signals for fine-tuning the base model $\mathcal{M}_0$ into the planning-enhanced model $\mathcal{M}_{P2E}$.

## 5.4 SUPERVISED FINE-TUNING

Following dataset construction, we conduct SFT to evolve the base model $\mathcal{M}_0$ into the planning-enhanced model $\mathcal{M}_{P2E}$. Each aligned reasoning tuple is defined as $\zeta_{align} = \langle \mathcal{Q}, \tau, \text{CoT}_\tau \rangle$, and the full training dataset is $\mathcal{C} = \{\zeta_{align}^i\}_{i=1}^N$.

During SFT, the model is trained to generate both the action sequence $\tau$ and the explanatory trajectory $\text{CoT}_\tau$ conditioned on the input problem $\mathcal{Q}$. The training objective is the standard autoregressive language modeling loss:

$$\mathcal{L}_{\text{SFT}} = -\sum_{i=1}^N \sum_{t=1}^{T_i} \log P(y_{i,t} \mid \mathcal{Q}_i; \theta), \tag{8}$$

where $T_i$ is the length of the supervised output sequence for the $i$-th instance, $y_{i,t}$ denotes the $t$-th token of the concatenated plan and reasoning trace $\langle \tau_i, \text{CoT}_{\tau_i} \rangle$, and $\theta$ are the model parameters.

This process compels the model to reproduce symbolically correct plans while simultaneously producing coherent reasoning chains. By learning from training dataset $\mathcal{C}$, PLAN2EVOLVE closes the gap between symbolic planning and language modeling, yielding $\mathcal{M}_{P2E}$ achieving greater planning accuracy, improved adaptability to novel tasks, and more reliable reasoning over extended horizons.

# 6 EXPERIMENTAL SETUP

## 6.1 DATA AND EVALUATION METRICS

We evaluate our approach on a variety of planning tasks. During evaluation, the model is prompted with the problem description alone, without additional information, prompting skills, or reasoning traces. Here is the information for training and testing dataset.

**Training Data:** We have adopted `Blocks World Classic (BW Classic)` comprises traditional stacking, unstacking, and reordering problems, from the Blocksworld benchmark, with optimal solution lengths normally distributed between 0 and 20 steps (Valmeekam et al., 2023; Liang et al., 2024). `Blocks World Hard (BW Hard)` increases difficulty by extending the planning horizon, featuring solution lengths uniformly distributed up to 60 steps. `Blocks World Align (BW Align)` introduces additional actions and orientation-related requirements, with solution lengths also uniformly distributed between 0 and 60 steps. We eventually obtained a training dataset size of 5807 for SFT. Details are in Section B.3.1.

**Testing Data:** In addition to the seen task types used in training, we also include unseen tasks, `Prepare Experiment`, `Reorganize Room`, and `Machine Parts Assembly`, during testing to evaluate the model's generalization and transferability to real-world scenarios. These tasks require similar actions but involve more diverse objects, environments, and goals. The solution lengths are uniformly distributed between 0 and 60 steps. In total, the test set contains over 300 previously unseen objects and 50 furniture types, forming 1,400 novel test experiments across four environments. This design enables a thorough evaluation of the model's ability to generalize. Further details are provided in Section B.2.1.

**Evaluation Metrics:** We adopt two evaluation metrics: 1) The **planning success rate** measures overall planning performance as the ratio of successfully completed tasks to the total number of tasks, following prior works (Garrett et al., 2020; Silver et al., 2023; Huang et al., 2025a). 2) The **progress score** quantifies the similarity between the goal state and the resulting state after the first invalid action. This metric is designed to capture partial correctness, particularly in super long-horizon tasks where LLMs rarely achieve full success. The details on calculating these two metrics are shown in Section B.2.4. The metric is formally defined as: given a planning problem $\mathcal{Q} = \langle \mathcal{O}, \mathcal{D}, \mathcal{X}^{(\text{init})}, \mathcal{X}^{(\text{goal})} \rangle$ and the model's predicted plan valid until $m^{th}$ step, $\tau_{valid} = \{a^{(0)}, \ldots, a^{(m)}\}, \mathcal{X}^{(\text{init})} \times \tau_{valid} \rightarrow \mathcal{X}^{(\text{pred})}$, $progress\_score = |\mathcal{X}^{(\text{goal})} \cap \mathcal{X}^{(\text{pred})}| / |\mathcal{X}^{(\text{goal})} \cup \mathcal{X}^{(\text{pred})}|$.

## 6.2 IMPLEMENTATION

The symbolic solver is based on the Fast-Forward (FF) planner (Hoffmann, 2001), with a Python interface provided by (Garrett et al., 2020). FF is a domain-independent symbolic planner that generates plans for arbitrary PDDL problems. Each training trace is defined as $\zeta^{\text{align}} = \langle \mathcal{Q}, \tau, \text{CoT}_\tau \rangle$, where $\mathcal{Q}$ is the planning problem, $\tau$ is the ground-truth plan, and $\text{CoT}_\tau$ is the aligning chain-of-thought. To prevent overfitting to narrow prompting styles, $\mathcal{Q}$ is dynamically paraphrased by the base model during data generation (Wang et al., 2023b). The fine-tuned model is referred to as PLAN2EVOLVE (P2E). Applying PLAN2EVOLVE to Qwen3-8B yields **P2E-8B**, while applying it to Qwen3-4B yields **P2E-4B** (Team, 2025b). Additional details are provided in Section B.3.

## 6.3 BASELINE AND ABLATION

**Baseline Models** We utilize several variants of the Qwen3 model family (Team, 2025b) as PLAN2EVOLVE base models, specifically Qwen3-4B-Instruct-2507 (denoted as Qwen3-4B) and Qwen3-8B. The resulting planning-enhanced models are compared not only against their original base models but also against other SOTA LLMs of similar sizes. These include open-source models such as Gemma3-4B and Gemma3-12B (Team, 2025a), as well as closed-source models such as

GPT-4o(OpenAI, 2023). Details on baseline model selection and implementation are provided in Section B.2.3.

**Baseline Approaches**  We also compare our approach against several reasoning-enhancing techniques commonly used to improve the planning ability of LLMs without requiring expert supervision. These include self-distillation, which refines the model using its own high-confidence outputs (Hinton et al., 2015); prompted Chain-of-Thought (Prompt-CoT) (Wei et al., 2022), where explicit reasoning prompts are provided to guide multi-step planning; and majority voting (Wang et al., 2023a), which aggregates multiple sampled trajectories to select the most consistent plan.

**Teacher-Student Ablation**  PLAN2EVOLVE is primarily designed as a self-teaching method, but it can also be applied in a teacher–student setting (P2ETS). To this end, we introduce an ablation of PLAN2EVOLVE, which leverages a teacher model to generate planning domain and symbolic-language aligned traces $\zeta^{align} = \langle \mathcal{Q}, \tau, \mathrm{CoT}_\tau \rangle$ for student model fine-tuning. In our experiments, we employ Qwen3-30B-A3B-Thinking (Qwen3-30B) (Team, 2025b) as the teacher model to fine-tune Qwen3-4B and Qwen3-8B. For naming, a Qwen3-4B model fine-tuned with Qwen3-30B as the teacher is denoted P2ETS-4B(30B), and similarly, the Qwen3-8B variant is denoted P2ETS-8B(30B). We further compare P2ETS against two baselines: (i) a base model fine-tuned directly on symbolic traces $\tau^{sym}$ without symbolic–language alignment, referred to as SymbOnly-4B (using Qwen3-4B as the base model), and (ii) a distillation baseline constructed by collecting direct responses from the teacher model to the problem statements, following the conventional distillation method (Hinton et al., 2015), denoted as Distill-4B(30B)/ Distill-8B(30B) (e.g., Qwen3-4B/Qwen3-8B distilled from Qwen3-30B).

**Reinforcement Learning Ablation**  Another ablation of PLAN2EVOLVE explores the incorporation of reinforcement-learning–based fine-tuning to determine whether RL can further enhance planning performance beyond supervised learning. Specifically, we fine-tune Qwen3-4B using Direct Preference Optimization (DPO) (Rafailov et al., 2023) and Constrained Policy Optimization (CPO) (Achiam et al., 2017), denoted as P2E-4B-DPO and P2E-4B-CPO, respectively. Additionally, we evaluate a hybrid variant in which CPO is applied on top of the supervised fine-tuned model P2E-4B, referred to as P2E-4B-SFT-CPO.

## 7 RESULT

In this section, we present our experimental results to address the following research questions: **RQ1.** Can PLAN2EVOLVE enable a base LLM to enhance its own planning capabilities? **RQ2.** How does the planning-enhanced model compare with SOTA models of similar size? **RQ3.** Does the enhanced model demonstrate stronger performance on unseen task types? **RQ4.** Can PLAN2EVOLVE advance thinking efficiency? **RQ5.** How does PLAN2EVOLVE compare to baseline planning enhancing approaches?

Table 1: Planning success rate and progress score across tasks for P2E and SOTA baselines of similar size. The best results are highlighted in bold, second best are underlined.

| Model | Seen Tasks Success Rate | | | Unseen Tasks Success Rate | | | Overall | |
|---|---|---|---|---|---|---|---|---|
| | BW Classic | BW Hard | BW Align | Prepare Experiment | Reorganize Room | Machine Parts Assembly | Success Rate | Progress Score |
| P2E-8B (ours) | **0.71** | **0.43** | **0.76** | 0.41 | **0.22** | 0.43 | **0.44** | **0.75** |
| Qwen3-8B | 0.48 | 0.28 | 0.69 | 0.33 | 0.19 | 0.40 | 0.35 | 0.68 |
| P2E-4B (ours) | 0.58 | 0.41 | 0.67 | **0.42** | 0.17 | **0.49** | 0.43 | 0.67 |
| Qwen3-4B | 0.41 | 0.24 | 0.42 | 0.24 | 0.12 | 0.34 | 0.26 | 0.59 |
| Gemma3-12B | 0.09 | 0.08 | 0.14 | 0.06 | 0.04 | 0.11 | 0.08 | 0.56 |
| Gemma3-4B | 0.01 | 0.01 | 0.01 | 0.01 | 0.01 | 0.01 | 0.01 | 0.44 |
| GPT-4o | 0.31 | 0.17 | 0.54 | 0.10 | 0.05 | 0.11 | 0.19 | 0.55 |

**RQ1 Effect of Plan2Evolve** The results indicate that the planning-enhanced models, P2E-8B and P2E-4B, substantially show improved planning capabilities compared to their base models. As shown in Table 1, both models achieve consistent gains in overall planning success rate and progress score

across seen and unseen tasks. P2E-8B achieves the strongest overall performance among all evaluated models, while P2E-4B yields the largest relative improvement, outperforming its base model by 17% and approaching the performance of Qwen3-8B, despite having only half the model size. On the challenging `BW Hard` benchmark, which requires reasoning over super long-horizon planning problems, P2E-8B and P2E-4B deliver notable gains of 15% and 17%, respectively, underscoring their enhanced reasoning abilities. The task-specific progress score can be found in Section B.4

**RQ2 Comparison to Other LLMs** We listed the top-performance open-source LLM baselines of similar size in Table 1. Results show that relative to other SOTA models, the planning-enhanced variants demonstrate clear superiority. P2E-8B and P2E-4B surpass similarly sized or even larger open-source models, by margins over 35%. They further outperform strong closed-source baseline GPT-4o, by as much as 24%.

**RQ3 Generalization** Beyond improvements on seen tasks, P2E-8B and P2E-4B also exhibit stronger generalization to unseen tasks, with performance gains ranging from 3% to 18% depending on the task, as shown in Table 1. The advantages observed in novel real-world scenarios, such as housekeeping, laboratory experiments, and factory production, indicate that the fine-tuned models have effectively internalized planning capabilities and enhanced their reasoning, rather than merely memorizing the training problems. These findings demonstrate the scalability and applicability of our method to a broad spectrum of robotic tasks.

**RQ4 Planning Efficiency** To assess planning efficiency, we analyze the ratio between the planning success rate and the token cost incurred during planning. As shown in Figure 2, all models fine-tuned under the PLAN2EVOLVE framework achieve both higher success rates and lower token costs compared to their respective base models, highlighting not only improved planning ability but also greater efficiency. Among them, P2ETS-8B offers the most favorable trade-off between performance and token usage. Notably, P2E-4B attains a higher success rate with lower token cost than Qwen-8B, despite being only half its size. These results demonstrate that PLAN2EVOLVE enables models to plan more efficiently. An detailed comparison of answers with different token lengths is provided in Section B.4.

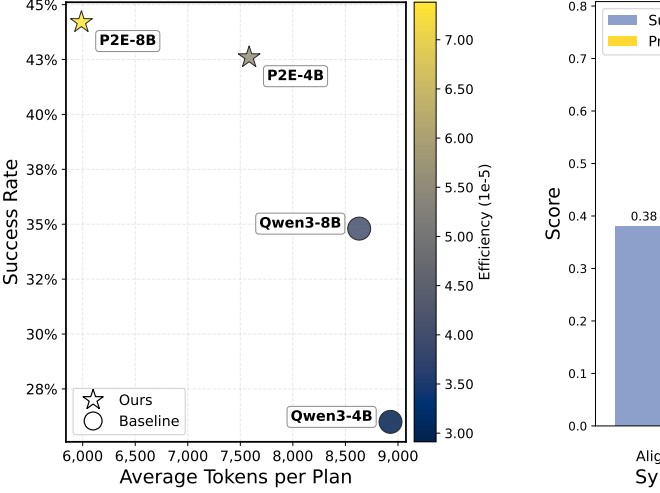

Figure 2: Token efficiency comparison between PLAN2EVOLVE models and baselines

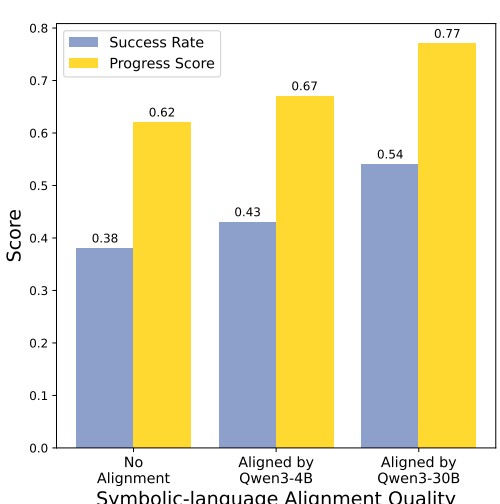

Figure 3: Overall planning performances under varied symbolic-language COT quality.

**RQ5 Comparison to Baseline Approaches** Table 2 shows that although these reasoning-enhancing baselines offer modest improvements over the raw Qwen3-4B model, none achieved the gains by PLAN2EVOLVE. P2E-4B delivers more than a 10% absolute boost in overall success rate and a substantially higher progress score. This demonstrates that P2E's structure-aware trajectories and domain-grounded plans provide a far stronger training signal than surface-level prompting strategies or heuristic self-refinement methods.

Table 2: Planning success rate and progress score across tasks for baselines approaches and PLAN2EVOLVE.The best results are highlighted in bold, second best are underlined.

| Model | Seen Tasks Success Rate | | | Unseen Tasks Success Rate | | | Overall | |
| | BW Classic | BW Hard | Align | Prepare Experiment | Reorganize Room | Machine Parts Assembly | Success Rate | Progress Score |
|---|---|---|---|---|---|---|---|---|
| P2E-4B (Ours) | **0.58** | **0.41** | **0.67** | **0.42** | **0.17** | **0.49** | **0.43** | **0.75** |
| Qwen3-4B (Majority Vote) | 0.46 | 0.26 | 0.49 | 0.30 | 0.15 | 0.39 | 0.32 | 0.66 |
| Qwen3-4B (Prompt-CoT) | 0.43 | 0.22 | 0.45 | 0.24 | 0.12 | 0.33 | 0.27 | 0.64 |
| Qwen3-4B (Self-Distill) | 0.45 | 0.23 | 0.44 | 0.25 | 0.13 | 0.35 | 0.28 | 0.62 |

Table 3: Planning success rate and progress score across tasks comparing PLAN2EVOLVE (teacher-student) and distillation. The best results are highlighted in bold, second best are underlined.

| Model | Seen Tasks Success Rate | | | Unseen Tasks Success Rate | | | Overall | |
| | BW Classic | BW Hard | BW Align | Prepare Experiment | Reorganize Room | Machine Parts Assembly | Success Rate | Progress Score |
|---|---|---|---|---|---|---|---|---|
| P2ETS-8B(30B) (ours) | **0.88** | **0.68** | 0.75 | **0.57** | **0.44** | **0.63** | **0.63** | **0.85** |
| P2ETS-4B(30B) (ours) | 0.71 | 0.52 | **0.81** | 0.52 | 0.28 | 0.58 | 0.54 | 0.77 |
| Distill-8B(30B) | 0.68 | 0.39 | 0.70 | 0.47 | 0.19 | 0.40 | 0.42 | 0.67 |
| Distill-4B(30B) | 0.50 | 0.31 | 0.74 | 0.23 | 0.16 | 0.49 | 0.36 | 0.54 |

# 8 ABLATION STUDY

In this section, we present three ablation studies of PLAN2EVOLVE to answer the following questions: **RQ6.** How critical is the role of symbolic–language alignment CoT, and how does its quality influence training outcomes? **RQ7.** Can PLAN2EVOLVE be effectively applied in a teacher–student setting, and does it outperform direct knowledge distillation? **RQ8.** Can reinforcement-learning–based techniques be integrated into PLAN2EVOLVE framework?

**RQ6 Symbolic-language Alignment Importance** We evaluate the impact of symbolic–language alignment by comparing the performance of models trained on alignment CoT data generated from different sources. To vary CoT quality, we generate symbolic–language alignments using models with different levels of reasoning capability. As shown in Figure 3, stronger teacher models provide higher-quality CoT, which in turn yields better student performance. Consequently, P2ETS-4B(30B), trained with alignment CoT generated by Qwen3-30B, achieves the highest overall success rate of 54% and the highest progress score of 77% among all trained models. Performance degrades progressively when the CoT is generated by weaker model Qwen3-4B. In contrast, when no alignment is performed, SymbOnly-4B, trained solely on symbolic plans and states, shows minimal improvement in overall planning performance and even degrades on some tasks, which is shown in Section B.4. These results highlight that symbolic–language alignment is crucial for training LLMs to plan with symbolic structures, and that the quality of alignment directly determines planning performances.

**RQ7 Teacher-Student Plan2Evolve** We evaluate the effectiveness of PLAN2EVOLVE in a teacher–student setting by analyzing the training outcomes of P2ETS models and comparing them with direct distillation, as shown in Table 3. The results demonstrate that PLAN2EVOLVE is a highly effective teacher–student training strategy as both P2ETS-8B(30B) and P2ETS-4B(30B) achieve clear improvements in planning success rate and progress score over their base models. Furthermore, PLAN2EVOLVE is proven to be superior to direct knowledge distillation, as P2ETS-8B(30B) outperforms Distill-8B(30B) by 21% in overall success rate and 18% in progress score, with similar trends observed for the 4B models. Notably, the progress score of P2ETS-8B(30B) even exceeds that of the teacher model Qwen3-30B by 9%, with details in Section B.4.

**RQ8 Reinforcement Learning within Plan2Evolve** We investigate whether PLAN2EVOLVE can effectively incorporate reinforcement-learning–based fine-tuning by evaluating two representative RL methods, including DPO and CPO, within our framework, as summarized in Table 4. Both P2E-DPO and P2E-4B-CPO improves overall success rate by over 3%, while P2E-4B-CPO achieves stronger benefits on structurally constrained tasks such as BW-Hard. Nevertheless, the advantage of supervised fine-tuning remains clear: P2E-SFT exceeding both RL variants by over 10%. A hybrid variant that further fine-tunes the SFT-trained model with RL (P2E-4B-SFT-CPO) achieves the strongest

Table 4: Planning success rate and progress score for SFT- and RL-based variants of PLAN2EVOLVE. The best results are highlighted in bold, second best are underlined.

| | Seen Tasks Success Rate | | | Unseen Tasks Success Rate | | | Overall | |
|---|---|---|---|---|---|---|---|---|
| **Model** | **BW Classic** | **BW Hard** | **BW Align** | **Prepare Experiment** | **Reorganize Room** | **Machine Parts Assembly** | **Success Rate** | **Progress Score** |
| P2E-4B-SFT-CPO (ours) | **0.60** | **0.45** | **0.75** | **0.45** | **0.18** | **0.50** | **0.46** | **0.84** |
| P2E-4B (ours) | 0.58 | 0.41 | 0.67 | 0.42 | 0.17 | 0.49 | 0.43 | 0.75 |
| P2E-4B-CPO (ours) | 0.52 | 0.33 | 0.52 | 0.29 | 0.17 | 0.35 | 0.31 | 0.69 |
| P2E-4B-DPO (ours) | 0.47 | 0.27 | 0.49 | 0.27 | 0.16 | 0.36 | 0.29 | 0.67 |
| Qwen3-4B (Baseline) | 0.41 | 0.24 | 0.42 | 0.24 | 0.12 | 0.34 | 0.26 | 0.59 |

performance across all metrics, raising the overall success rate to 0.46 and the progress score to 0.84, showing that RL serves as an effective refinement mechanism once symbolic–language alignment has been established by SFT.

## 9 REAL ROBOT EXPERIMENT

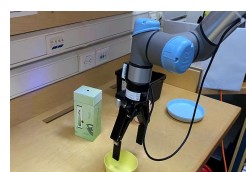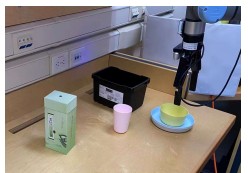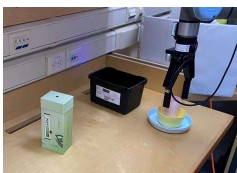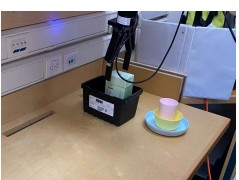

Figure 4: P2E-4B performing room organization task together with $\pi_0$

We deploy P2E-4B on a UR5e real robot, using $\pi_0$ (Black et al., 2024) as the motion planning skill, to perform the Reorganize Room task. The task-plan outputs are directly used as prompts for $\pi_0$. This experiment demonstrates both the deployability of our method in real-world robotic systems and its compatibility with state-of-the-art manipulation algorithms. Details are in Section B.7

## 10 LIMITATIONS

While PLAN2EVOLVE demonstrates strong improvements in planning performance, several limitations remain. First, the framework requires the base LLM to have sufficient reasoning and generation capability to produce valid PDDL planning domains. This prerequisite limits its applicability to small models, which often lack the capacity to reliably generate coherent symbolic domains. Second, although training with symbolic–language alignment substantially improves LLMs' planning performance, its quality remains bounded by its own reasoning ability to align symbolic plan. Our current approach does not directly improve the symbol-language alignment quality of the model, leaving room for future methods that can jointly enhance capabilities in planning and alignment.

## 11 CONCLUSION

In this work, we introduced PLAN2EVOLVE, a self-evolving framework that leverages LLM-generated planning domains as scalable sources of verified supervision. By treating domains as data engines, PLAN2EVOLVE automatically constructs diverse, long-horizon problem–plan pairs that surpass the base model's intrinsic planning ability without human curation. Through self-symbolic–language alignment CoT, it bridges formal symbolic structures with natural-language reasoning, internalizing search behavior and planning skills. Empirically, models fine-tuned with PLAN2EVOLVE achieve large gains in planning success, generalization, and token efficiency. We further show that symbolic–language alignment is crucial for training LLMs to plan, and that PLAN2EVOLVE is effective both in self-evolving and teacher–student training. Future work will scale PLAN2EVOLVE to broader applications, incorporate multi-modal inputs, and develop methods that jointly enhance symbolic-language alignment quality and planning capability, and explore multi-iteration approach to further bootstrap the planning performance.

## 12 REPRODUCIBILITY STATEMENT

We have made substantial efforts to ensure the reproducibility of our work. The core PLAN2EVOLVE framework is implemented in Python, with all key algorithms detailed in Algorithm 1. Planning domain generation method is shown in Section B.1, with details of symbolic planner and robot simulators. All datasets used for evaluation and training can be reproduced following Section B.1 with their exact composition and difficulty distributions detailed in Section B.2.1 and Section B.3.1. Our evaluation pipeline, including the symbolic planning integration, testing prompt template, and metrics computation, is fully specified in Section B.2.3. The training methodology, including hyperparameters, data processing steps, and training prompt templates, are comprehensively documented in Section B.3.

## 13 ETHICS STATEMENT

We have carefully read and affirm our commitment to the principles outlined in the ICLR Code of Ethics, including honesty, fairness, transparency, and the avoidance of harm. All methodological details, data sources, and limitations have been thoroughly documented to promote reproducibility and accountability. No sensitive personal or private data were used in this work.

We do not foresee any immediate risks or potential misuses arising from the proposed methods. Our intention is for this work to contribute to scientific progress while remaining aligned with ethical standards and community values.

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

# A  THE USE OF LARGE LANGUAGE MODELS (LLMS)

This paper is focusing on the study of LLM training, in which large language models are the primary object of investigation and a diverse range of models are evaluated through experiments.

Beyond their role as the target of study, LLMs were employed as tools for grammar correction and stylistic refinement, including improvements in clarity and readability. LLMs were not involved in the generation of research ideas, the design or implementation of experiments, the analysis of data, or the interpretation of findings. The authors have full responsibility for the originality, accuracy, and integrity of all scientific content reported in this work.

# B  APPENDIX

## B.1  LLM-BASED DOMAIN GENERATION

### B.1.1  INITIAL DOMAIN SKELETON CONSTRUCTION

The first stage of PLAN2EVOLVE is the automatic construction of symbolic planning domains. We leverage the generative capacity of the base model $\mathcal{M}_0$ to propose candidate predicates that capture object relations and intrinsic properties, guided by physical simulation. The overall pipeline follows the domain generation framework proposed by Huang et al. (2025b). Input $\mathcal{U}$ is a demonstration trajectory with a short task description. Subsequently, we prompt $\mathcal{M}_0$ with demonstration trajectories collected by the Agilex Pika Data Collection System to invent actions, which are then compiled into executable planning domains. The model outputs a preliminary PDDL domain skeleton $\mathcal{D}_0 = \langle \mathcal{P}, \mathcal{A} \rangle$, where $\mathcal{P}$ denotes the set of predicates and $\mathcal{A}$ the set of actions with preconditions and effects. This skeleton is then tested against a suite of sampled planning problems $\{\mathcal{Q}_i\}$ using the Fast-Forward planner (Hoffmann, 2001; Garrett et al., 2020). Successful execution indicates a consistent domain; otherwise, domain errors are detected and used for refinement.

### B.1.2  FEEDBACK–DRIVEN PLANNING DOMAIN REPAIR

When validation fails, PLAN2EVOLVE introduces two complementary self-correction mechanisms.

**Feedback prompting.**  Error traces from the planner (e.g.,undefined predicate) are reformulated into feedback prompts. These prompts are re-injected into $\mathcal{M}_0$ to request targeted corrections, following the iterative refinement proposed in (Guan et al., 2023a; Oswald et al., 2025).

Formally, given error $e$ produced on problem $\mathcal{Q}_i$, we define a feedback function

$$h(e, \mathcal{Q}_i) \rightarrow \text{diagnostic prompt } d,$$

which is appended to the domain-fix query to produce a repaired domain $\mathcal{D}_{t+1}$. This iteration is repeated until a consistent and executable domain $\mathcal{D}'$ is generated. Prompt Template is as following:

---

**Prompt for Domain Error Fixing**

### **Role** ###
You are an expert in AI Planning (PDDL) and robotics task modeling. Your task is to fix mistakes of a PDDL planning domain.

### **PDDL Domain** ###
The current domain is:{Current domain}

### **Problem** ###
The planning problem is: {Planning Problem}

### **Error** ###
An error occurred during solving planning problem, the returned error is:{Error Trace}

---

**Hill-Climbing Search for Domain Redundancy Pruning**  In addition to error repair, generated domains often contain redundant predicates and actions. To address this, we employ a symbolic

hill-climbing algorithm (Silver et al., 2023; Kumar et al., 2023; Huang et al., 2025a) that prunes unnecessary components from the domain. This procedure ensures that the final domain $\mathcal{D}^\star$ is both executable and minimal, containing only semantically necessary components.

### B.1.3 AUTOMATIC PROBLEM–PLAN PAIR GENERATION

Once a validated domain $\mathcal{D}^\star$ is obtained, we generate a library of problem–plan pairs $(\mathcal{Q}, \tau)$. Each problem is constructed by sampling initial and goal states consistent with $\mathcal{D}^\star$, and solved with the symbolic planner. The resulting pairs are later aligned with chain-of-thought explanations to form the training traces used in PLAN2EVOLVE.

## B.2 EVALUATION DETAILS

### B.2.1 EVALUATION DATA DETAILS

The evaluation dataset consists of seven disjoint task datasets: *stack-200*, *unstack-200*, *reorder-200*, and *align-200* (Blocks World domain), along with *prepare-experiment-200*, *reorganize-room-200*, and *machine-parts-assembly-200*. Each dataset comprises 200 tasks, with solution lengths uniformly sampled across four intervals: 0–10, 10–20, 20–30, and 30+ steps.

Next, we discuss how each testing set is augmented and the detailed difficulty distribution:

**Seen Tasks:**

- *Blocks World-Classic* is a reproduction of the traditional Blocks World benchmark, consisting of 100 problem instances across the stack, unstack, and reorder tasks. The optimal plan lengths approximately follow a normal distribution within 0–20 steps. The details of test problem distribution of Blocks World Classic is shown in Figure 5

- *Blocks World-Hard* extends the benchmark to include more challenging problems with longer optimal plan lengths up to 60 steps. The distribution of problem counts across four difficulty intervals is kept balanced. It contains 200 problem instances for each of the three task types: stack, unstack, and reorder.

- *Blocks World-Align* further extends the benchmark by introducing orientation reasoning. In addition to the standard actions, a rotate action is included, requiring the model to reason about spatial orientations.

**Unseen Tasks**

- *Reorganize Room*: The robot must collect household items, redistribute them to their designated locations, and pack them according to specified requirements.

- *Machine Parts Assembly*: The robot must collect machining parts distributed across the factory and assemble them in the required order.

- *Prepare Experiment*: The robot must retrieve laboratory equipment and set up an experimental platform.

The unseen tasks include a large scale diverse objects (over 300) and furniture(over 50), the details are shown in the following section.

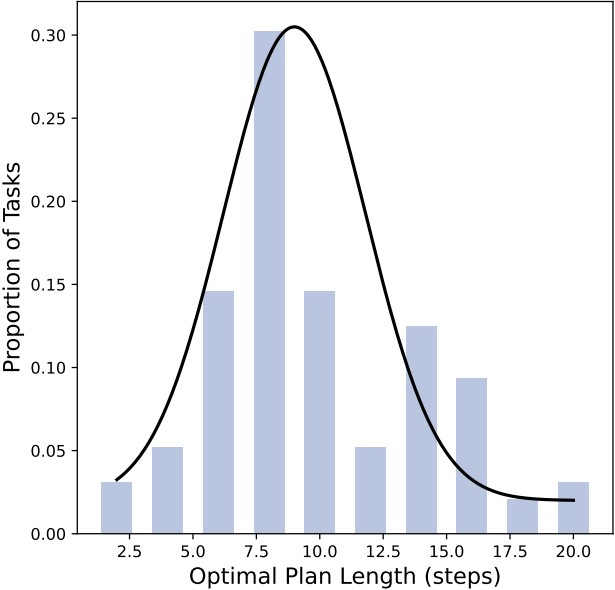

Figure 5: Evaluation data distribution for Blocks World Classic

B.2.2 Unseen Object Types and Furniture Included in Test Set

**HouseKeeping Objects**

shoebox, book, towel, cushion, pillow, blanket, toyblock, jar, canister, bin, basket, tilepack, box, storagebox, detergent, soapbar, tissuebox, magazine, photoalbum, cuttingboard, foodbox, ricebag, flourbag, sugarbag, spicejar, candle, cup, plate, pot, pan, tray, bucket, stepbox, organizer, toybox, craftbox, sewingbox, pillowbox, laundrybox, clothbag, storagecrate, hamper, cushionbox, shelfbox, matpack, drivecase, clipboard, penbox, pencilbox, markerbox, staplebox, tape, tapeholder, calendar, planner, report, documentbox, letterbox, envelopebox, badgebox, tagbox, cardbox, stampbox, inkpad, paperroll, chartbook, whiteboard, pinboard, notepad, scrapbook, catalog, supplybox, lunchbox, laptopbox, headsetcase, monitorbox, keyboardbox, mousebox, cablebox, dockbox, shoes, slippers, sandals, boots, books, magazines, notebooks, comics, albums, photoalbums, towels, napkins, blankets, pillows, cushions, plates, bowls, cups, glasses, mugs, cutlery, forks, spoons, knives, chopsticks, spicejars, condiments, cerealboxes, snackpacks, bottles, jars, cans, storagebins, shoeboxes, laundrybaskets, soapbars, detergents, shampoos, conditioners, lotions, toothbrushes, toothpastes, razors, combs, brushes, hats, scarves, belts, ties, gloves

**Factory Objects**

pallet, crate, ingot, brick, block, mold, drum, barrel, tray, spool, battery, foam, plate, rod, beam, sheet, coil, carton, gearbox, motor, casing, bearingbox, brickpack, cablebox, metalbox, plasticbin, boltpail, nutbox, washerbox, pipebundle, timber, lumber, steelbar, rebar, partbox, panel, duct, filterbox, container, powderbag, sack, clampbox, toolkit, spacerblock, fastenerbox, weldrod, fixture, drillbox, pallets, crates, bricks, blocks, beams, pipes, rods, bars, rebars, sheets, panels, plates, coils, rolls, cylinders, drums, barrels, containers, boxes, cartons, bolts, nuts, washers, screws, clamps, wrenches, spanners, drills, toolbits, sockets, filters, gaskets, valves, hoses, cables, chains, belts, wheels, gears, motors, casings, bearings, molds, fixtures, frames, foampads, straps, seals, packaging, labels

**Lab Objects**

rack, cylinder, labbox, carton, container, samplebox, tipbox, cryobox, pack, dish, slidebox, capsule, pouch, filterbox, tray, case, testbox, bufferbox, kit, bag, tubecrate, platebox, mediumbottle, sealbag, gelbox, reagentbox, chipbox, cellbox, rackbox, capbox, powderjar, acidbottle, solventcan, stockbottle, samplejar, drybox, packtube, enzymebox, coolerbox, chemcart, bottles, beakers, flasks, cylinders, vials, tubes, testtubes, petri, slides, racks, tipboxes, cryoboxes, samplebags, pipettes, pipettips, dishes, capsules, ampoules, filters, funnels, gloves, masks, goggles, aprons, coats, notebooks, pens, labels, markers, tags, trays, cases, carts, stands, supports, boxes, containers, jars, pouches, packs, media, solutions, buffers, reagents, kits, cells, chips, plates, serums, enzymes

**Housekeeping Furniture**

dining table, coffee table, side table, console table, end table, bedside table, kitchen table, foldable table, picnic table, patio table, round table, square table, rectangular table, buffet table, sofa table, low table, tea table, serving table, bench table, counter table, island table, tv stand, hall table, display table, exhibit table, study desk, writing desk, computer desk, standing desk, reception desk, conference table, meeting table, office table, printer stand, workstation, drafting table, blueprint table

> **Factory Furniture**
>
> workbench, assembly table, packing table, utility table, sorting table, assembly bench, grinding table

> **Lab Furniture**
>
> lab bench, lab table, specimen table, experiment bench, fume table, inspection table

### B.2.3 EVALUATION IMPLEMENTATION DETAILS

We built a unified evaluation pipeline for all experiments. The pipeline loads each evaluation dataset and constructs prompts by combining a system prompt with a task-specific user prompt template. For model inference, a maximum generation length of up to 16,384 tokens are allowed. The tokenizer's built-in chat template is applied to each prompt to ensure consistent formatting. For each model output we extract the final predicted action sequence enclosed in `<FINAL>` tags.

> **System Prompt for Evaluation**
>
> You are a robot assistant. Your task is to generate a plan given the initial and goal state. A plan is a sequence of actions.

> **User Prompt for Evaluation**
>
> ### **General request###**
> Your task is to predict a set of actions that arrive at the goal state starting the initial state. A state is defined by a set of predicates. Predicates can be static (i.e. describe invariant properties of the environment that do not change over time) or dynamic.
>
> ### **Possible Predicates ###** : {Your Predicates}
>
> ### **Possible Actions ###**: {Your Actions}
>
> ### **Problem to Solve ###**: {Initial State} {Goal State}
>
> ### **Output ###**:Always output the final plan inside `<FINAL>` ... `</FINAL>`

> **Code for Extracting Final Action Sequence**
>
> ```python
> def extract_answer(output):
>     # This pattern ensures no nested <FINAL> inside the capture
>     matches = re.findall(r'<FINAL>((?:(?!<FINAL>).)*?)</FINAL>',
>         output, re.DOTALL)
>     if matches:
>         return re.sub(r'([^,\[\]\s]+)', r'"\1"', matches[-1])  #
>             last match only
>     return None
> ```

### B.2.4 EVALUATION METRIC DETAILS

We evaluated our models in principle on two metrics: **Success Rate** and **Progress Score**. The formal definitions of the metrics follow:

Given a planning problem,

$$\mathcal{Q} = \langle \mathcal{O}, \mathcal{D}, \mathcal{X}^{(\text{init})}, \mathcal{X}^{(\text{goal})} \rangle,$$

and the model's prediction,

$$\tau = \{a^{(0)}, \ldots, a^{(T-1)}\},$$

we define,
$$\mathcal{X}_N^{(\text{plan})} = \mathcal{X}^{(\text{init})} \times a^{(0)} \times \cdots \times a^{(N-1)}.$$

Furthermore, we define that for any action $a^{(i)} \notin$ action space of $\mathcal{X}$ (which is an invalid action),
$$\mathcal{X} \times a^{(i)} = \emptyset,$$

and for all $j$,
$$\emptyset \times a^{(j)} = \emptyset.$$

Thus, if the model's predicted plan is valid until $m^{th}$ step, it follows that,
$$m = \min\big(\{\, i \mid \mathcal{X}^{(\text{init})} \times a^{(0)} \times \cdots \times a^{(i)} = \emptyset \,\} \cup \{T\}\big).$$

We define the logical divergence function to describe similarity between 2 states,
$$f_{\text{logical divergence}}(\mathcal{X}^{(i)}, \mathcal{X}^{(j)}) = \frac{|\mathcal{X}^{(i)} \cap \mathcal{X}^{(j)}|}{|\mathcal{X}^{(i)} \cup \mathcal{X}^{(j)}|}.$$

Finally,
$$\textbf{Success Rate}(\mathcal{Q}, \tau) = \mathbf{1}\Big[\mathcal{X}_T^{(\text{plan})} = \mathcal{X}^{(\text{Goal})}\Big] \tag{9}$$

$$\textbf{Progress Score}(\mathcal{Q}, \tau) = f_{\text{logical divergence}}(\mathcal{X}_m^{(\text{plan})}, \mathcal{X}^{(\text{goal})}) \tag{10}$$

## B.3 TRAINING DETAILS

### B.3.1 TRAINING DATA DETAILS

The training problems are randomly sampled from the generated PDDL domain, with solution lengths ranging from 0 to 60 steps, resulting in a total of 5,807 examples. Among these, 719 are from `Blocks World Align`, 3,048 from `Blocks World Hard`, and 2,038 from `Blocks World Reorder`. For `Blocks World Hard`, the optimal plan lengths follow a 6:2:1:1 ratio across the intervals 0–10, 10–20, 20–30, and 30+ steps. The symbolic solver is permitted to generate both optimal and suboptimal solutions, allowing the model to learn from shortest-path plans as well as alternative, longer trajectories. During training, an evaluation set is held out, consisting of 80, 340, and 227 examples for the respective task types, corresponding to an evaluation ratio of 0.1.

Each training example, consisting of a problem definition and its corresponding solution (either optimal or suboptimal), is provided to the LLM, which then generates a symbolic-language chain-of-thought alignment. These alignments, together with the problems and solutions, form the ground truth of the training dataset. The system prompt used during training is identical to that used in evaluation. Additionally, we allow dynamic rephrasing of the problem setting in the user prompt during training (relative to evaluation) to help the model maintain focus on the problem context. The details of training problem distribution is shown in Figure 6.

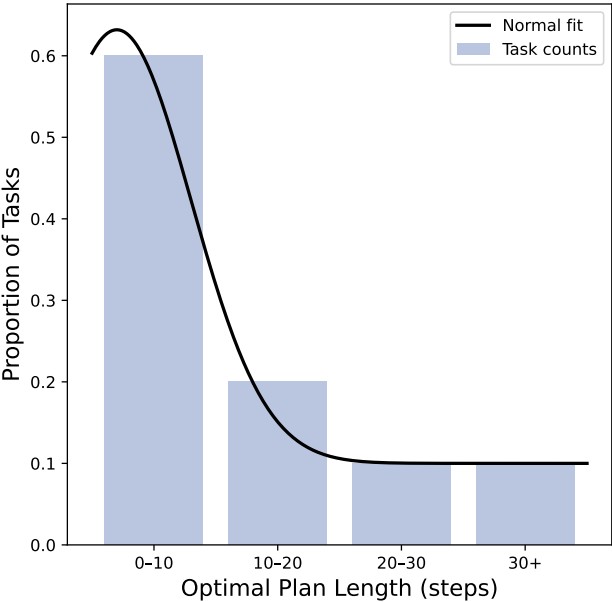

Figure 6: Evaluation data distribution for Blocks World Classic

### B.3.2 TRAINING IMPLEMENTATION DETAILS

The pipeline for generating CoT follows a structure similar to the evaluation pipeline, with minor modifications to the prompts and a maximum generation length of 65,536 tokens. From each model output, we extract the final predicted action sequence enclosed within `<REASON>` tags. During training, we set the learning rate to $1 \times 10^{-5}$, trained for 5 epochs, used a per-device batch size of 2, and applied gradient accumulation over 2 steps. Optimizer adopted is Adam.

### System Prompt for Generating Symbolic-language alignment CoT

### **Role** ###
You are an expert in AI Planning (PDDL) and robotics task modeling. Your task is to generate a detailed chain-of-thought reasoning process for solving the given planning problem.

### **Goal** ###
You will be provided with:
- The planning domain
- The initial state
- The goal state
- The ground truth task plan

### **Task Description** ###
Your job is to produce a step-by-step reasoning process that explains:
- Why each action was chosen
- How each action changes the state
- How the evolving state satisfies preconditions and leads toward the goal
- The logical connections between actions, state transitions, and goal achievement
- Explore a few applicable actions at each step other than the provided ground truth
- **After each step, briefly reflect on why alternative actions were not chosen at that point**

### **Output** ###
You can follow the EXAMPLE reasoning provided, return the full result.

### User Prompt for Generating Symbolic-language alignment CoT

### **Problem Setting**: ### {Your Problem Setting}

### **Task Description** Your task is to explain how to predict a set of actions that arrive at the goal state starting the initial state.

### **Task to explain**: ###
Initial State: {Your Initial State}
Goal State: {Your Goal State}
Correct plan of actions: {Your Symbolic Plan}

### **Solution**: ### After your reasoning, put your final explanation in this format:
{<REASON><ANSWER_HERE></REASON> }

### Code for Extracting CoT Response

```python
def extract_answer(output):
    # This pattern ensures no nested <FINAL> inside the capture
    matches = re.findall(r'<REASON>((?:(?!<REASON>).)*?)</REASON>',
        output, re.DOTALL)
    if matches:
        return matches[-1]  # last match only
    return None
```

## B.4 MORE EXPERIMENT RESULTS

Here we provide the detailed success rate, progress score, and average tokens used of each evaluated model under very task type, including some not listed in main body.

Table 5: Planning success rate across tasks for all evaluated models.

| Model | Seen Tasks Success Rate | | | Unseen Tasks Success Rate | | | Overall |
|---|---|---|---|---|---|---|---|
| | BW Classic | BW Hard | BW Align | Prepare Experiment | Reorganize Room | Machine Parts Assembly | Success Rate |
| P2E-8B | 0.71 | 0.43 | 0.76 | 0.41 | 0.22 | 0.43 | 0.44 |
| P2E-4B | 0.58 | 0.41 | 0.67 | 0.42 | 0.17 | 0.49 | 0.43 |
| SymbOnly-4B | 0.54 | 0.34 | 0.84 | 0.16 | 0.14 | 0.50 | 0.38 |
| P2ETS-8B(30B) | 0.88 | 0.68 | 0.75 | 0.57 | 0.44 | 0.63 | 0.63 |
| Distill-8B(30B) | 0.68 | 0.39 | 0.70 | 0.47 | 0.19 | 0.40 | 0.42 |
| P2ETS-4B(30B) | 0.71 | 0.52 | 0.81 | 0.52 | 0.28 | 0.58 | 0.54 |
| Distill-4B(30B) | 0.50 | 0.31 | 0.74 | 0.23 | 0.16 | 0.49 | 0.36 |
| P2E-4B-SFT-CPO | 0.60 | 0.45 | 0.75 | 0.45 | 0.18 | 0.50 | 0.46 |
| P2E-4B-CPO | 0.52 | 0.33 | 0.52 | 0.29 | 0.17 | 0.35 | 0.31 |
| P2E-4B-DPO | 0.47 | 0.27 | 0.49 | 0.27 | 0.16 | 0.36 | 0.29 |
| Qwen3-30B | 0.96 | 0.70 | 0.82 | 0.84 | 0.47 | 0.82 | 0.72 |
| Qwen3-8B | 0.48 | 0.28 | 0.69 | 0.33 | 0.19 | 0.40 | 0.35 |
| Qwen3-4B | 0.41 | 0.24 | 0.42 | 0.24 | 0.12 | 0.34 | 0.26 |
| Qwen3-1.7B | 0.07 | 0.04 | 0.01 | 0.00 | 0.00 | 0.00 | 0.02 |
| Qwen2.5-7B | 0.02 | 0.01 | 0.01 | 0.00 | 0.00 | 0.00 | 0.00 |
| Gemma-3-4b | 0.01 | 0.01 | 0.01 | 0.00 | 0.01 | 0.01 | 0.01 |
| Gemma-3-12b | 0.09 | 0.08 | 0.14 | 0.06 | 0.04 | 0.11 | 0.08 |
| GPT-4o | 0.31 | 0.17 | 0.54 | 0.10 | 0.05 | 0.11 | 0.19 |

Table 6: Planning progress score across tasks for all evaluated models.

| Model | Seen Tasks Progress Score | | | Unseen Tasks Progress Score | | | Overall |
|---|---|---|---|---|---|---|---|
| | BW Classic | BW Hard | BW Align | Prepare Experiment | Reorganize Room | Machine Parts Assembly | Progress Score |
| P2E-8B | 0.93 | 0.72 | 0.95 | 0.68 | 0.62 | 0.81 | 0.75 |
| P2E-4B | 0.80 | 0.63 | 0.91 | 0.66 | 0.51 | 0.74 | 0.67 |
| SymbOnly-4B | 0.79 | 0.58 | 0.98 | 0.44 | 0.46 | 0.71 | 0.62 |
| P2ETS-8B(30B) | 0.93 | 0.86 | 0.93 | 0.80 | 0.72 | 0.89 | 0.85 |
| Distill-8B(30B) | 0.82 | 0.60 | 0.92 | 0.66 | 0.55 | 0.73 | 0.67 |
| P2ETS-4B(30B) | 0.86 | 0.75 | 0.97 | 0.75 | 0.59 | 0.85 | 0.77 |
| Distill-4B(30B) | 0.71 | 0.46 | 0.92 | 0.41 | 0.40 | 0.68 | 0.54 |
| P2E-4B-SFT-CPO | 0.94 | 0.76 | 0.95 | 0.70 | 0.65 | 0.84 | 0.76 |
| P2E-4B-CPO | 0.85 | 0.66 | 0.88 | 0.61 | 0.59 | 0.74 | 0.69 |
| P2E-4B-DPO | 0.82 | 0.63 | 0.86 | 0.59 | 0.58 | 0.72 | 0.67 |
| Qwen3-30B | 0.95 | 0.72 | 0.88 | 0.89 | 0.56 | 0.84 | 0.76 |
| Qwen3-8B | 0.78 | 0.62 | 0.93 | 0.64 | 0.57 | 0.72 | 0.68 |
| Qwen3-4B | 0.73 | 0.58 | 0.72 | 0.52 | 0.52 | 0.63 | 0.59 |
| Qwen3-1.7B | 0.58 | 0.46 | 0.70 | 0.37 | 0.46 | 0.40 | 0.47 |
| Qwen2.5-7B | 0.51 | 0.46 | 0.75 | 0.43 | 0.49 | 0.45 | 0.50 |
| Gemma-3-4b | 0.51 | 0.43 | 0.69 | 0.34 | 0.41 | 0.35 | 0.44 |
| Gemma-3-12b | 0.64 | 0.53 | 0.83 | 0.49 | 0.50 | 0.53 | 0.56 |
| GPT-4o | 0.68 | 0.54 | 0.88 | 0.41 | 0.45 | 0.49 | 0.55 |

Table 7: Planning token count across tasks for all evaluated models.

| Model | Seen Tasks Token Count | | | Unseen Tasks Token Count | | | Overall |
|---|---|---|---|---|---|---|---|
| | BW Classic | BW Hard | BW Align | Prepare Experiment | Reorganize Room | Machine Parts Assembly | Token Count |
| P2E-8B | 5060 | 6338 | 4595 | 5864 | 6811 | 5632 | 5988 |
| P2E-4B | 5536 | 8448 | 5391 | 7686 | 8348 | 6316 | 7584 |
| SymbOnly-4B | 7689 | 10328 | 4483 | 10876 | 9908 | 7494 | 9107 |
| P2ETS-8B(30B) | 5380 | 6432 | 5483 | 6411 | 7478 | 5879 | 6364 |
| Distill-8B(30B) | 9349 | 11358 | 7850 | 10293 | 11170 | 9843 | 10462 |
| P2ETS-4B(30B) | 4263 | 6163 | 3063 | 5240 | 7178 | 4702 | 5524 |
| Distill-4B(30B) | 9101 | 10785 | 5558 | 10624 | 10788 | 7560 | 9555 |
| P2E-4B-SFT-CPO | 5521 | 8437 | 5298 | 7654 | 8312 | 6341 | 7543 |
| P2E-4B-CPO | 5042 | 6345 | 4650 | 5887 | 6795 | 5610 | 6012 |
| P2E-4B-DPO | 5610 | 8572 | 5413 | 7792 | 8390 | 6203 | 7621 |
| Qwen3-30B | 9435 | 11418 | 9612 | 9948 | 11923 | 10339 | 10868 |
| Qwen3-8B | 8564 | 9368 | 6409 | 8413 | 8983 | 8510 | 8631 |
| Qwen3-4B | 8409 | 9963 | 6657 | 8319 | 9323 | 8316 | 8929 |
| Qwen3-1.7B | 7133 | 7440 | 6518 | 7268 | 6980 | 5917 | 7000 |
| Qwen2.5-7B | 563 | 905 | 560 | 1165 | 570 | 550 | 794 |
| Gemma-3-4b | 765 | 926 | 772 | 953 | 772 | 909 | 883 |
| Gemma-3-12b | 582 | 704 | 563 | 732 | 550 | 617 | 653 |
| GPT-4o | 2590 | 2761 | 2790 | 2791 | 2640 | 2796 | 2757 |

## B.5 OVERALL PIPELINE PSEUDO CODE

---

**Algorithm 1** Full Planning Pipeline

---

1: **procedure** FULLPLANPIPELINE($O, M_0, \Psi, U, N$)
2: $\quad (\hat{P}, \hat{A}) \leftarrow \Psi_{M_0}(U)$ $\hfill \triangleright$ Generate predicates/actions
3: $\quad \hat{D} \leftarrow (\hat{P}, \hat{A})$ $\hfill \triangleright$ Construct domain
4: $\quad \mathcal{C} \leftarrow \emptyset$ $\hfill \triangleright$ Problem–CoT pairs
5: $\quad$ **for** $i = 1$ to $N$ **do**
6: $\quad\quad (X^{init}, X^{goal}) \leftarrow$ RANDOMSAMPLE$(O, \hat{D})$
7: $\quad\quad Q_i \leftarrow (O, \hat{D}, X^{init}, X^{goal})$
8: $\quad\quad \tau \leftarrow$ PDDL_SOLVER$(Q_i)$
9: $\quad\quad$ CoT$_\tau \leftarrow \{f_{M_0}^{NL}(X^t, a^t, X^{t+1})\}_{t=0}^{T-1}$
10: $\quad\quad \mathcal{C} \leftarrow \mathcal{C} \cup \{(Q_i, \text{CoT}_\tau)\}$
11: $\quad$ **end for**
12: $\quad M_{P2E} \leftarrow$ TRAIN$(M_0, \mathcal{C})$ $\hfill \triangleright$ Abstract training step
13: $\quad$ Scores $\leftarrow \emptyset$
14: $\quad$ **for** $Q_j \in$ TestSet **do**
15: $\quad\quad \hat{y} \leftarrow M_{P2E}(Q_j)$
16: $\quad\quad \hat{\tau} \leftarrow$ EXTRACTPLAN$(\hat{y})$
17: $\quad\quad$ Scores $\leftarrow$ Scores $\cup \{$PLANVALIDATION$(\hat{\tau}, Q_j)\}$
18: $\quad$ **end for**
19: $\quad$ Score$_{avg} \leftarrow \frac{1}{|\text{Scores}|} \sum$ Scores
20: $\quad$ **return** $M_{P2E}$, Score$_{avg}$
21: **end procedure**

---

## B.6 EXAMPLE LLM-GENERATED PDDL DOMAIN

---

**System Prompt for Generating CoT**

```
(define (domain LLM_generaated_domain)
    (:requirements :strips :equality)
    (:predicates
        (obj ?b1)
        (on-table ?b1 ?t1)
        (holding ?b1 ?r1)
        (hand_free ?r1)
        (top ?b2)
        (above ?b1 ?b2)
        (robot ?r1)
        (table ?t1)
        (alined ?b1 ?b2)
    )

    (:action pick-up
        :parameters (?b1 ?t1 ?r1)
        :precondition (and (obj ?b1) (robot ?r1)(on-table ?b1
        ?t1) (top ?b1) (hand_free ?r1) (table ?t1))
        :effect (and (not (hand_free ?r1)) (not (on-table ?b1
        ?t1)) (holding ?b1 ?r1))
    )

    (:action stack
        :parameters (?b1 ?b2 ?r1)
        :precondition (and (obj ?b1)(top ?b1) (holding ?b1 ?r1)
        (robot ?r1)(top ?b2) (obj ?b2))
        :effect (and (above ?b1 ?b2) (hand_free ?r1) (not (top
        ?b2)) (not (holding ?b1 ?r1)))
    )

    (:action unstack
        :parameters (?b1 ?b2 ?r1)
        :precondition (and (obj ?b1) (robot ?r1)(top ?b1)
        (above ?b1 ?b2) (hand_free ?r1)(obj ?b2))
        :effect (and (not (hand_free ?r1)) (not (above ?b1
        ?b2)) (top ?b2) (holding ?b1 ?r1))
    )

    (:action put-down
        :parameters (?b1 ?t1 ?r1)
        :precondition (and (obj ?b1)(holding ?b1 ?r1) (top ?b1)
        (robot ?r1) (table ?t1))
        :effect (and (hand_free ?r1) (not (holding ?b1 ?r1))
        (on-table ?b1 ?t1))
    )

    (:action rotate
        :parameters (?b1 ?b2 ?t1 ?r1)
        :precondition (and (table ?t1) (robot ?r1) (obj ?b2)
        (obj ?b1) (on-table ?b2 ?t1) (holding ?b1 ?r1))
        :effect (and (alined ?b1 ?b2))
    )
)
```

## B.7 VLA Implementation Details

We integrate our P2E-4B model as a high-level planner with the $\pi_0$ model (Black et al., 2024), which serves as the low-level executor, to control a UR5e robot in a set of table-organization tasks within our real-robot experiment.

The P2E-4B model produces high-level commands expressed in PDDL, which are passed to the $\pi_0$ model. The $\pi_0$ model, fine-tuned on 200 real-world pick-and-place trajectories using PDDL-based prompts, translates these plans into low-level spatial delta poses for the UR5e. Execution is then handled by the UR5e's built-in controllers.

The overall task involves picking up and placing objects in different locations to place everything in place and reach a required table-top configuration. This VLA experiment serves as a proof of concept, demonstrating that our high-level planners can be seamlessly connected to a lower-level VLA model to carry out real-world tasks.

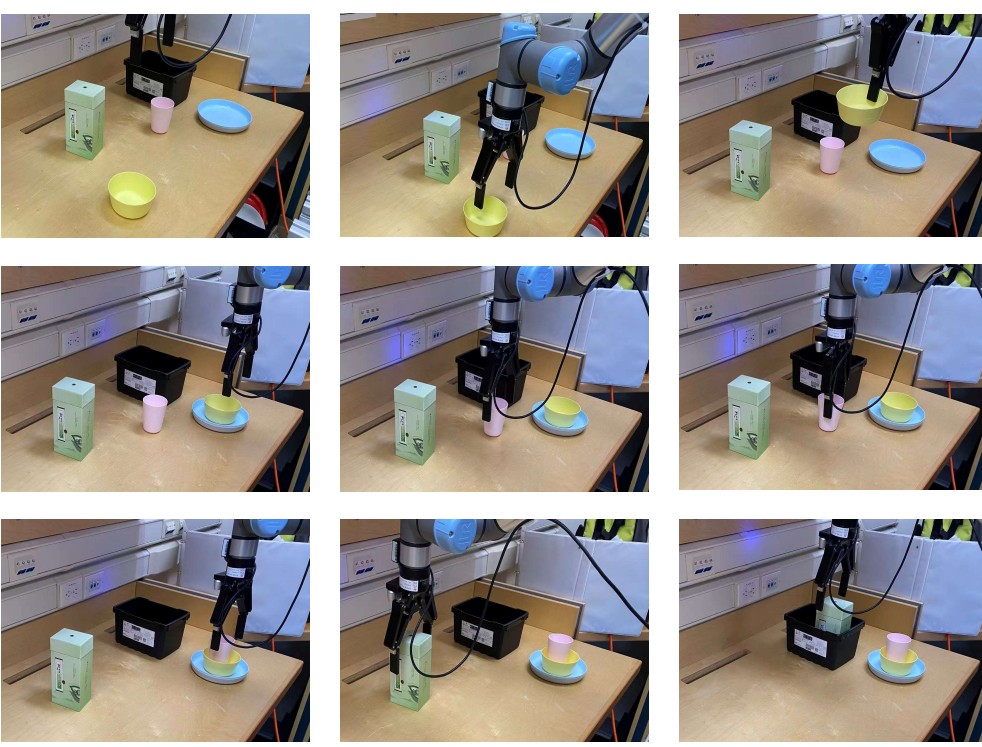

Figure 7: P2E-4B performing room organization task together with $\pi_0$

## B.8 BACKBONE ABLATION

To assess the generality of Plan2Evolve (P2E) beyond the Qwen family, we applied the full pipeline to Llama-3.1-8B, which differs substantially from Qwen in both architecture and training recipe. Integrating P2E yields substantial improvements in both success rate and progress score, indicating gains not only in task completion but also in planning quality for partially solved tasks. While the improvements on unseen tasks are smaller, P2E still provides consistent benefits. These cross-model results show that P2E is not tied to Qwen and can effectively strengthen a broad range of LLMs, including those that start with weaker symbolic-planning capabilities.

Table 8: Model planning success rate comparing before/after fine-tuning on Llama model.

| Model | Seen Tasks Success Rate | | | Unseen Tasks Success Rate | | | Overall |
| | BW Classic | BW Hard | Align | Prepare Experiment | Reorganize Room | Machine Parts Assembly | Success Rate |
| --- | --- | --- | --- | --- | --- | --- | --- |
| Llama-3.1-8B | 0.005 | 0.003 | 0.013 | 0.000 | 0.000 | 0.000 | 0.003 |
| Llama-3.1-8B-P2E(Ours) | **0.330** | **0.184** | **0.135** | 0.000 | 0.000 | 0.000 | **0.098** |

Table 9: Model progress score comparing before/after fine-tuning on Llama model.

| Model | Seen Tasks Progress Score | | | Unseen Tasks Progress Score | | | Overall |
| | BW Classic | BW Hard | Align | Prepare Experiment | Reorganize Room | Machine Parts Assembly | Progress Score |
| --- | --- | --- | --- | --- | --- | --- | --- |
| Llama-3.1-8B | 0.538 | 0.463 | 0.800 | 0.447 | 0.484 | 0.462 | 0.512 |
| Llama-3.1-8B-P2E(Ours) | **0.742** | **0.610** | **0.847** | 0.448 | 0.485 | 0.461 | **0.580** |

