# OpenReview forum: "Plan2Evolve: LLM Self-Evolution for Improved Planning Capability via Automated Domain Generation"
_ICLR.cc/2026/Conference — Submitted to ICLR 2026_

### Official Review · Reviewer_8Nk8 · 2025-10-27

**Soundness:** 3
**Presentation:** 3
**Contribution:** 3
**Rating:** 6
**Confidence:** 4

**Summary:**

This paper introduces Plan2Evolve, a self-evolving framework designed to enhance the task planning capabilities of Large Language Models (LLMs) by generating and learning from Chain-of-Thought (CoT) data. The approach bridges two key research areas: (a) the use of LLMs for Planning Domain Definition Language (PDDL) domain generation, and (b) the enhancement of CoT reasoning in LLMs.

This is achieved by repurposing LLM-generated PDDL domains as a source of scalable and verifiable training data. The process involves several stages: First, the framework first employs the base LLM to generate and self-correct a valid PDDL domain. Then, a symbolic solver is used to produce verified problem-plan pairs. Nextly, these symbolic pairs are converted to natural language pairs, forming CoT reasoning trajectories. Finally, the model is fine-tuned on these data.

Experimental results demonstrate that this method significantly improves the model's planning success rate, its ability to generalize to unseen tasks, and its overall efficiency. The study also underscores the critical importance of symbolic-language alignment and proves the framework's effectiveness compared to direct knowledge distillation.

The core contributions lie in demonstrating that an LLM's planning capabilities can enhance planning capabilities by: (1) self-generating scalable and verifiable data; (2) leveraging structured knowledge from tools like PDDL solvers; and (3) aligning formal symbolic structures with natural language reasoning.

**Strengths:**

1. The proposed method reduces the need for expensive, human-curated planning datasets, and enables models to achieve performance that surpasses even larger baseline models.

2. The approach is straightforward, effective, and highly reproducible.

3. The paper situates its work by identifying two relevant research areas—symbolic planning and Chain-of-Thought (CoT) post-training for LLMs. A key contribution is its novel approach to bridging these two fields.

4. The experiment is detailed, including comparisons and ablation studies that demonstrate the effectiveness of the proposed method.

**Weaknesses:**

Please refer to the question section.

**Questions:**

# On the Limitations of the PDDL-based Approach

1.1 The PDDL framework has inherent limitations in its expressiveness and cannot encompass the full spectrum of real-world planning tasks. How might this constrain the broader applicability and scalability of the Plan2Evolve framework?

1.2 The method relies on a symbolic solver to generate the valid plans that form the CoT data. This raises two concerns: First, could the LLM's learning process be seen as merely imitating the solver's logic? Second, since many real-world problems are not readily solvable by existing symbolic planners, does this reliance on a solver create a performance ceiling for Plan2Evolve?

1.3 The training dataset is solely based on the Blocks World task. Does this narrow training domain potentially limit the model's generalization capabilities on more diverse, unseen tasks? Have the authors considered whether expanding the variety of tasks in the training set could lead to further performance improvements?

# On the "Self-Evolving" Nature of the Framework

2.1 As noted in Q1.2, if the core planning logic is derived from an external symbolic solver, does the LLM primarily function as a translator between symbolic and natural language representations? This raises the question of whether the performance gains stem from the solver's embedded knowledge rather than a truely continuous "self-evolution" of the LLM's intrinsic planning abilities.

2.2 Line 424 highlights that the quality of symbolic-language alignment is critical for effective training. How does the model's own alignment capability change after undergoing SFT with Plan2Evolve? It would be insightful to see an evaluation of the model's text-to-symbol alignment performance (e.g., on benchmarks like [1, 2]) both before and after fine-tuning.

2.3 Could the fine-tuned Plan2Evolve model be used to generate a new, potentially higher-quality dataset for a subsequent round of training? Would an iterative, online learning-style approach, where the model continuously refines its own training data, lead to sustained performance improvements?

Clarification and additional experiments on these points would be very helpful in strengthening the paper.

*[1] TEXT2WORLD: Benchmarking Large Language Models for Symbolic World Model Generation. Hu et al.*

*[2] PROC2PDDL: Open-Domain Planning Representations from Text. Zhang et al.*

---

> ### Author Response · Authors · 2025-12-03
> **Reply to Q1.1-1.3**
>
> We thank the reviewer for the positive and constructive feedback. We appreciate the recognition of our method’s effectiveness, reproducibility, and ability to achieve strong performance without costly human-curated datasets. Your feedback is very thoughtful and constructive. Below, we address your concerns in detail.
>
> ## Q1.1
>
> We agree that the limited expressiveness of PDDL is a common challenge in the field. However, Plan2Evolve (P2E) is designed to learn planning behavior, not to memorize fixed PDDL plans. Our experiments on unseen tasks demonstrate consistent improvement, indicating that the model internalizes the underlying planning patterns rather than relying on specific domain encodings.
>
> Importantly, the trained LLM does not inherit the brittleness of PDDL, where even small omissions or minor specification errors in a domain/problem file can cause complete failure. As long as the high-level description of the environment and goals remains close, the model continues to function robustly. This robustness is further evidenced by our practice of dynamically varying the prompting structure and rephrasing problem statements during training and evaluation—the model remains stable despite these changes.
>
> Thus, while PDDL’s formal expressiveness has limits, Plan2Evolve’s learned planning behavior enables broader applicability and better scalability than a purely symbolic planner tied strictly to PDDL definitions. Besides, we also experimentally show that the Plan2Evolve can be intergated with generalizable low-level controllers to generalize to complex manipulation tasks, as indicated in Line 447-461.
>
>
> As a future direction, we believe the model can further generalize beyond the solver’s capabilities by incorporating additional modalities (e.g., visual cues or continuous feedback) that enhance the expressiveness beyond PDDL.
>
> ## Q1.2
>
> One of our central goals is for the LLM to internalize planning logic and behavior rather than memorize specific solutions, similar to how a search algorithm generalizes beyond individual solved instances. In addition, we explicitly leverage the LLM’s commonsense knowledge to generalize beyond the coverage of a fixed PDDL domain. This is evidenced by the model’s strong performance on unseen tasks, where standard PDDL pipelines would typically require careful manual domain engineering.
>
> Regarding the potential performance ceiling: although using a symbolic solver nominally introduces an upper bound, in practice the bottleneck is not the solver but the LLM’s ability to internalize and generalize the demonstrated planning behavior. Our experiments support this observation: none of the tested SOTA LLMs were able to solve all tasks that a PDDL solver could handle, indicating that the solver is not the limiting factor. Once the model learns the underlying planning patterns, its performance is no longer tied to the solver’s coverage. It can extrapolate beyond the original demonstrations and even solve task variations that the original PDDL domain would fail on, as demonstrated by experiments to unseen tasks.
>
> ## Q1.3
>
> We agree that incorporating more training data and domains would likely improve performance. However, we intentionally restrict training to a Blocksworld dataset, as our goal is not for the model to memorize specific plans. Instead, we aim to test whether it can internalize fundamental planning behavior and transfer it to novel tasks, such as house-tidying and other unseen scenarios. Our results prove that, by learning from a simple, well-structured setting, the model can better acquire generalizable logical patterns and planning strategies applicable across diverse problem domains.

---

> ### Author Response · Authors · 2025-12-03
> **Reply to Q2.1-2.2**
>
> ## Q2.1
> We believe this concern is unlikely to hold for two key reasons grounded in how our training data is constructed.
>
> First, our training data consists exclusively of (problem description → chain-of-thought solution) pairs. We do not include any explicit symbolic–to–natural-language paired examples. The symbolic solver in our pipeline is used only to generate skeleton trajectories, and these raw symbolic outputs are never included in the training set. Only the chain-of-thought explanations written fully in natural language are used for finetuning. As a result, the model is never exposed to symbolic knowledge in a form that would teach it to translate between symbolic and natural-language representations. This makes it unlikely that the performance gains stem merely from improved translation.
>
> Second, if the model were only improving its translation ability, it would not account for the gains we observe on novel tasks and on unseen object–predicate combinations. These settings require the model to reason over new symbolic structures, generate consistent logical sequences, and construct entirely new long-horizon plans, but not to translate familiar tokens. Yet the model continues to improve on correct multi-step solutions even in these out-of-distribution scenarios. This strongly suggests that the model has internalized planning behavior, not symbolic decoding rules.
>
> Taken together, these points indicate that Plan2Evolve meaningfully enhances the model’s intrinsic planning competence, rather than functioning as a translation layer around an external solver.
>
> ## Q2.2
>
> We believe that Plan2Evolve is unlikely to substantially alter the model’s symbolic–language alignment capability, for two main reasons.
>
> (1) The training objective does not supervise symbolic–language alignment.
> Our SFT dataset does not provide paired symbolic and natural-language examples or reward signals that explicitly encourage better alignment. As a result, the fine-tuning process does not directly optimize the model for improved symbolic grounding or translation.
>
> (2) Bootstrapping with the fine-tuned model does not yield measurable gains.
> We experimented with regenerating chain-of-thought explanations using the fine-tuned model and using these to create a second round of SFT. This “self-refinement” did not noticeably improve performance, suggesting that the model’s alignment between symbolic plans and natural-language reasoning did not meaningfully change after the first round of fine-tuning.
>
> Taken together, these observations indicate that Plan2Evolve primarily improves the model’s planning competence, not its symbolic–language alignment. Thus, while benchmarks that evaluate symbolic–language mapping (e.g., [1, 2]) are valuable in other settings, they are not aligned with the training objective of our framework and would not be expected to show meaningful differences before and after SFT.
>
> Additionally, these benchmarks assess a different direction of mapping than what Plan2Evolve uses. Existing evaluations focus on natural language to symbolic translation, which can be quantitatively measured by comparing the predicted symbolic description to ground truth. In contrast, our method operates in the reverse direction (symbolic plan → natural-language explanation). Current benchmarks do not provide a principled or standardized way to evaluate how well a natural-language explanation corresponds to a symbolic definition or plan, making it difficult to measure the alignment quality.
>
> We agree that not jointly improving alignment quality is a limitation, and we have highlighted this as an important direction for future work in our discussion.

---

> ### Author Response · Authors · 2025-12-03
> **Reply to Q2.3**
>
> ## Q2.3
>
> We investigated this idea empirically. In principle, one could use the fine-tuned Plan2Evolve model to generate new (problem → CoT solution) pairs and then conduct another training round on this “self-generated” dataset. However, in our experiments, a straightforward second-round training did not produce noticeable performance gains. The self-generated data largely remained within the same distribution and exhibited similar quality to the original data, so simply amplifying it did not meaningfully improve planning competence. In particular, modest improvements in CoT quality did not appear sufficient to bootstrap stronger planning behavior. As shown in the , fine-tuning Qwen3-4B on CoT generated by Qwen3-8B showed no clear improvement, whereas using substantially higher-quality CoT from Qwen3-32B did help. This suggests that the CoT quality must be significantly higher to yield observable benefits.
> #### Table R1. Success rate of P2E model trained using CoT of different quality
> | Training Setup                               | Success Rate |
> |--------------------------------------------|--------------|
> | P2E-4B trained with Qwen-4B CoT           | 43%          |
> | Qwen-4B trained with Qwen-8B CoT           | 44%          |
> | Qwen-4B trained with Qwen-30B CoT          | 54%          |
>
>
> That said, we consider this a promising direction for future work. Iterative or online-style refinement may become effective if paired with mechanisms that (i) actively improve CoT quality (e.g., through filtering, ranking, or preference-based selection), or (ii) intentionally shift the training distribution toward harder, more diverse, or systematically perturbed problems rather than reusing similar trajectories. Developing such a principled curriculum or data-selection scheme for Plan2Evolve is an exciting avenue we hope to explore in future iterations of the framework.

---

### Official Review · Reviewer_NQ8s · 2025-10-27

**Soundness:** 2
**Presentation:** 3
**Contribution:** 2
**Rating:** 4
**Confidence:** 3

**Summary:**

This work proposes a new method to enhance the planning capacity of LLMs. Specifically, an LLM is employed to generate symbolic representations in the form of Planning Domain Definition Language (PDDL) from natural language task descriptions. The resulting planning domain and task solution obtained from a PDDL solver are then translated back into natural language with chains of thought. These translated examples collectively form a dataset used for supervised fine-tuning to strengthen the planning capability of the LLM.

The paper is grounded in the observation that natural language explanations articulated through chains of thought can more effectively enhance the planning capacity of LLMs than fine-tuning directly on raw symbolic solutions. The curated training data translated from symbolic transitions capture stepwise symbolic reasoning patterns that often exceed the innate planning ability of LLMs. Overall, the proposed method is conceptually sound and easy to follow. Empirical results provide partial evidence of its advantage over LLMs that are not fine-tuned.

**Strengths:**

1. This work demonstrates that training on natural language translated from symbolic solutions can enhance the planning capacity of LLMs. In generating the planning domain, the authors design a two-stage procedure to produce an executable PDDL domain. During the symbolic-language alignment stage, they construct well-designed prompts from four perspectives to ensure that the generated language contains not just a linear narration of the symbolic solution but rich reasoning steps that enable the LLM to understand the task and its solution more holistically. These technical designs are commendable.

2. In the experimental section, the authors evaluate not only the planning capability within the designed environments but also the generalization performance on unseen tasks, which to some extent strengthens the contribution of this paper.

**Weaknesses:**

1. In the experimental section, the authors select only unfine-tuned LLMs as baseline methods and do not compare their approach with other LLM-based methods specifically designed for planning tasks. From my perspective, this is the main limitation of the work. Although the proposed method shows noticeable improvements, it remains unclear how much it contributes relative to existing approaches in the community. A more convincing evaluation would demonstrate that this method also outperforms prior fine-tuned LLMs developed for planning tasks.

2. The proposed method is implemented only on Qwen and not transferred to other base LLMs such as Llama or GPT-4o-mini. I suspect that some LLMs may lack the ability to generate executable planning domains solely from task descriptions. Nonetheless, it would still be valuable to see results showing how this method performs when built upon other base models.

3. The method is evaluated only on tasks with solution lengths of up to 60 steps. The paper does not examine its planning capability on longer-horizon tasks, which limits understanding of its scalability and robustness.

**Questions:**

1. Is it possible to fine-tune the LLM using RL instead of SFT within your proposed framework?

2. Do you have any plans to extend your model to other planning environments, such as RLBench [1], to further strengthen the contribution of your work?

[1] RLBench: The Robot Learning Benchmark & Learning Environment, RAL 2020.

---

> ### Author Response · Authors · 2025-12-02
> **Reply to W1**
>
> We sincerely thank the reviewer for the thoughtful feedback and for recognizing the strengths of our approach, including the symbolic-language alignment design and the evaluation on both in-domain and unseen tasks. Here we provide detailed explaination to address your concerns.
>
> ## W1
>
> The reviewer notes that the experiments only compare against unfine-tuned LLMs and do not benchmark against other LLM-based planning approaches. However, we did not find prior self-training methods specifically for robotic planning. To address this concern, we further evaluate Plan2Evolve against several widely adopted LLM reasoning-enhancement techniques. Specifically, we include: (1) Prompted Chain-of-Thought (Prompt-CoT) [1], which injects a detailed worked-out reasoning example into the prompt to guide the model toward step-by-step thinking; (2) Self-Distillation [2], where the model first produces higher-quality intermediate rationales and then learns from these generated explanations through supervised refinement; and (3) Majority Vote Ensembles [3], which generate multiple candidate plans and select the most consistent or frequently occurring one to reduce randomness and improve reliability. Under the no–online-supervision assumption, we exclude methods that rely on system feedback, as our goal is to evaluate the model’s planning ability in a single attempt. Thus, techniques such as failure-feedback are not considered. The selected baselines represent the most common and effective reasoning-enhancement strategies in recent LLM-planning research. Crucially, all of them operate without access to domain labels or structured executable plans, making them directly comparable to Plan2Evolve (P2E).
>
> The results in table R1.1 and R1.2 show that while these baselines provide minimal to moderate improvements over the raw model, none approach the gains achieved by P2E. A possible explanation is that for tasks requiring many sequential steps, prompt-based methods may lose focus, and majority vote can be diluted by conflicting solution candidates. This confirms that the structure-aware trajectories and domain-grounded plans generated by Plan2Evolve provide a fundamentally stronger training signal than surface-level prompting or self-refinement heuristics.
>
> #### Table R1.1. Model planning success rate compared against baselines.
> | Model                    | BW (Classic) | BW (Hard) | Align | Prepare Experiment | Reorganize Room | Machine Parts Assembly | Overall |
> | ------------------------ | ---------------- | ------------- | ----- | ------------ | --------------- | ---------------------- | ------- |
> | Qwen3-4B (Baseline)      | 0.41             | 0.24          | 0.42  | 0.24         | 0.12            | 0.34                   | 0.26    |
> | Qwen3-4B(Prompt-CoT)     | 0.43             | 0.22          | 0.45  | 0.24         | 0.12            | 0.33                   | 0.27    |
> | Qwen3-4B (Self-Distill)  | 0.45             | 0.23          | 0.44  | 0.25         | 0.13            | 0.35                   | 0.28    |
> | Qwen3-4B (Majority Vote) | 0.46             | 0.26          | 0.49  | 0.30         | 0.15            | 0.39                   | 0.32    |
> | Qwen3-4B-P2E             | 0.58             | 0.41          | 0.67  | 0.42         | 0.17            | 0.49                   | 0.43    |
>
> #### Table R1.2. Model planning progress score compared against baselines.
> | Model                     | BW (Classic) | BW (Hard) | Align | Prepare Experiment | Reorganize Room | Machine Parts Assembly | Overall |
> |---------------------------|------------------|---------------|-------|--------------|-----------------|------------------------|---------|
> | Qwen3-4B (Baseline)       | 0.73 | 0.58 | 0.72 | 0.52 | 0.52 | 0.63 | 0.59 |
> | Qwen3-4B(Prompt-CoT)                  | 0.82 | 0.63 | 0.78 | 0.56 | 0.55 | 0.67 | 0.64 |
> | Qwen3-4B (Self-Distill)            | 0.77 | 0.61 | 0.82 | 0.53 | 0.49 | 0.64 | 0.62 |
> | Qwen3-4B (Majority Vote)  | 0.80 | 0.65 | 0.87 | 0.55 | 0.58 | 0.71 | 0.66 |
> | Qwen3-4B-P2E (Ours)          | 0.93 | 0.72 | 0.95 | 0.68 | 0.62 | 0.81 | 0.75 |
>
> We included the above discussion and result in L427-440 of the revised manuscript.

---

> ### Author Response · Authors · 2025-12-02
> **Reply to W2-3**
>
> ---
> ## W2
>
> We acknowledge in the limitations section that our approach relies on the LLM’s ability to generate correct planning domains. We believe the method remains valuable given the rapid progress of LLMs, as models are becoming increasingly capable in both logical reasoning and structured code generation.
>
> Moreover, even when a model lacks sufficient ability to write planning domains, our experiments demonstrate that Plan2Evolve can still function effectively in a teacher–student setting (Table 2, L437). By using a stronger teacher model to generate planning domains, the weaker student achieves significant performance gains, highlighting that teacher-student variants of our method offer a practical and robust alternative, further mitigating this limitation.
>
> To assess the generality of Plan2Evolve (P2E) beyond the Qwen family, we applied the full pipeline to Llama-3.1-8B, which differs substantially from Qwen in both architecture and training recipe. Integrating P2E yields substantial improvements in both success rate and progress score, indicating gains not only in task completion but also in planning quality for partially solved tasks. While the improvements on unseen tasks are smaller, P2E still provides consistent benefits. These cross-model results show that P2E is not tied to Qwen and can effectively strengthen a broad range of LLMs, including those that start with weaker symbolic-planning capabilities.
>
> #### Table R2.1 Model planning success rate comparing before/after fine-tuning on Llama model
> | Model | BW (Classic) | BW (Hard) | Align | Prepare Experiment | Reorganize Room | Machine Parts Assembly | Overall |
> |-------|------------------|---------------|-------|--------------|-----------------|----------------|---------|
> | Llama-3.1-8B | 0.005 | 0.003 | 0.013 | 0.000 | 0.000 | 0.000 | 0.003 |
> | Llama-3.1-8B-P2E | 0.330 | 0.184 | 0.135 | 0.000 | 0.000 | 0.000 | 0.098 |
>
> #### Table R2.2 Model progress score comparing before/after fine-tuning on Llama model
> | Model            | BW (Classic) | BW (Hard) | Align | Prepare Experiment | Reorganize Room | Machine Parts Assembly | Overall |
> | ---------------- | ------------ | --------- | ----- | ------------ | --------------- | ---------------------- | ------- |
> | Llama-3.1-8B     | 0.538        | 0.463     | 0.800 | 0.447        | 0.484           | 0.462                  | 0.512   |
> | Llama-3.1-8B-P2E | 0.742        | 0.610     | 0.847 | 0.448        | 0.485           | 0.461                  | 0.580   |
>
> We included the above discussion and result in Appendix B.8 (L1512) of the revised manuscript.
>
> ---
>
> ## W3
>
> A 60-step horizon refers to high-level symbolic actions, not low-level motion frames. In TAMP, this is already considered very long: most established benchmarks (e.g., LIBERO [4], Blocksworld) involve fewer than 15 high-level actions. A 60-step plan corresponds to roughly 60 sequential pick/place/align operations, each requiring its own geometric feasibility check. Evaluating plans of this length therefore places the method squarely in a long-horizon, high-complexity regime where many existing systems begin to struggle. The following table reports the model’s success rate on tasks requiring 30+ high-level steps.
>
> #### Table R3.1 Model planning success rate on 30+ steps planning tasks
> | Model | BW (>30 steps) |
> |-------|------------------|
> |  Qwen3-4B (Baseline) | 0.13 |
> |  Qwen3-4B-P2E (Ours)  | 0.18 |
> |  Qwen3-4B-P2ETS(30B) (ours)   | 0.25 |
> |  Qwen3-30B (Baseline)  | 0.41 |
>
> #### Table R3.2 Model planning progress score on 30+ steps planning tasks
> | Model                     | BW (>30 steps) |
> |---------------------------|--------------------|
> | Qwen3-4B (baseline)       | 0.23               |
> | Qwen3-4B-P2E (ours)       | 0.36               |
> | Qwen3-4B-P2E-TS (30B)     | 0.55               |
> | Qwen3-30B (baseline)      | 0.62               |
>
>
> The results show that all models struggle on tasks longer than 30 steps. This indicates that including extremely long tasks is less informative, as they provide limited discriminative results when every model performs poorly. Therefore, our chosen task-length distribution is reasonable and appropriate for meaningful evaluation.

---

> ### Author Response · Authors · 2025-12-03
> **Reply to Q1-2 & References**
>
> ## Q1
> Yes. RLHF or environment-reward fine-tuning can naturally integrate into Plan2Evolve. We have explicitly evaluated two representative RL alignment methods (DPO [5] and CPO [6]) within Plan2Evolve. As shown in the tables, both P2E-DPO and P2E-CPO outperform the baseline, confirming that RL-based optimization can be seamlessly integrated into our pipeline. Among the two, CPO provides slightly stronger improvements, particularly on more structurally constrained tasks.
>
> That said, our results show that SFT remains the most effective single-stage training strategy for Plan2Evolve. A plausible explanation is that our P2E-generated dataset is relatively small compared to standard RLHF-scale corpora, making dense supervised signals more data-efficient than preference- or constraint-based RL. Consequently, P2E-SFT consistently surpasses both P2E-DPO and P2E-CPO in success rate and progress score.
>
> We also evaluate a hybrid variant that applies SFT followed by RL fine-tuning (P2E-SFT-CPO), which achieves the strongest overall performance. This suggests that RL effectively complements SFT by further refining plan validity and constraint satisfaction once the model has been aligned to the generated domains.
>
> # Applying Plan2Evolve using RL
>
> #### Table R4.1. Model planning success rate for SFT-and-RL-based approaches.
> | Model | BW (Classic) | BW (Hard) | Align | Prepare Experiment | Reorganize Room | Machine Parts Assembly | Overall |
> |-------|------------------|---------------|-------|--------------|-----------------|------------------------|---------|
> | Qwen3-4B (Baseline) | 0.41 | 0.24 | 0.42 | 0.24 | 0.12 | 0.34 | 0.26 |
> | Qwen3-4B-P2E-SFT | 0.58 | 0.41 | 0.67 | 0.42 | 0.17 | 0.49 | 0.43 |
> | Qwen3-4B-P2E-DPO | 0.47 | 0.27 | 0.49 | 0.27 | 0.16 | 0.36 | 0.29 |
> | Qwen3-4B-P2E-CPO | 0.52 | 0.33 | 0.52 | 0.29 | 0.17 | 0.35 | 0.31 |
> | Qwen3-4B-P2E-SFT-CPO | 0.60 | 0.45 | 0.75 | 0.45 | 0.18 | 0.50 | 0.46 |
>
> #### Table R4.2. Model planning progress score for SFT-and-RL-based approaches.
> | Model | BW (Classic) | BW (Hard) | Align | Prepare Experiment | Reorganize Room | Machine Parts Assembly | Overall |
> |-------|------------------|---------------|-------|--------------|-----------------|------------------------|---------|
> | Qwen3-4B (Baseline) | 0.73 | 0.58 | 0.72 | 0.52 | 0.52 | 0.63 | 0.59 |
> | Qwen3-4B-P2E-SFT | 0.93 | 0.72 | 0.9''5 | 0.68 | 0.62 | 0.81 | 0.75 |
> | Qwen3-4B-P2E-DPO | 0.82 | 0.63 | 0.86 | 0.59 | 0.58 | 0.72 | 0.67 |
> | Qwen3-4B-P2E-CPO | 0.85 | 0.66 | 0.88 | 0.61 | 0.59 | 0.74 | 0.69 |
> | Qwen3-4B-P2E-SFT-CPO | 0.94 | 0.76 | 0.95 | 0.70 | 0.65 | 0.84 | 0.76 |
>
> We included the above discussion and result in L480-500 of the revised manuscript.
>
> ---
>
> ## Q2
> We have carefully reviewed the RLBench setup. Unfortunately, we are not able to directly evaluate our method on RLBench because the environment provides feedback only in continuous space (e.g., image observations, joint states, distances to the goal). In contrast, Plan2Evolve plans and requires feedback in a logical state space (e.g., predicates such as is_on, next_to). We cannot directly let our models play in the RLBench environment without low-level controllers. Meanwhile, RLBench does not include functionality for mapping its continuous observations into symbolic predicates, so a direct comparison or evaluation is not currently feasible.
>
> Nevertheless, we hope that the RL-based results we provided help address your concern regarding the applicability of our approach across different settings.
>
> ---
>
> ### References
> [1] Wei et al. Chain-of-Thought Prompting Elicits Reasoning in Large Language Models, 2022.
>
> [2] Hinton et al. Distilling the Knowledge in a Neural Network, 2014.
>
> [3] Wang et al. Self-Consistency Improves Chain-of-Thought Reasoning in Language Models, 2023
>
> [4] Liu et al. LIBERO: Benchmarking Knowledge Transfer for Lifelong Robot Learning, 2023
>
> [5] Rafailov et al. Direct Preference Optimization: Your Language Model is Secretly a Reward Model, 2023.
>
> [6] Achiam et al. Constrained Policy Optimization, 2017.

---

### Official Review · Reviewer_3hhe · 2025-10-28

**Soundness:** 3
**Presentation:** 2
**Contribution:** 1
**Rating:** 2
**Confidence:** 5

**Summary:**

This paper presents Plan2Evolve, a framework for improving the planning capabilities of LLMs. The methods uses LLM-generated (using a strong LLM) symbolic planning domains (in PDDL), which are then used to produce problem/plan pairs, and then convert these symbolic traces into CoT explanations using the same model. The generated symbolic and natural-language traces form a training dataset for supervised fine-tuning (SFT), producing a specialized LLM.

Empirically, the paper shows that Plan2Evolve improves planning accuracy, cross-task generalization, and token efficiency on several benchmarks (Blocksworld variants and real-world robotic scenarios). Ablation studies analyze different choices for some of the parameters (e.g., teacher model).

**Strengths:**

- Both areas, (PDDL) planning and self-improvement of LLMs, are gathering attention recently, so the topic certainly fits the scope of ICLR
- The experimental results give solid results

**Weaknesses:**

(1) My main critique of the paper is that it is proposed as a self-evolving technique while it is not the case. There are several reasons why this is an overreaching claim:

- While the paper presents Plan2Evolve as a “self-evolving” framework, the experiments only show a single generation+fine-tuning iteration. The model is never shown to re-generate improved data using the evolved version. As such, the claimed self-evolving loop remains hypothetical rather than empirically validated. Demonstrating iterative self-improvement would have been a crucial step to substantiate the “self-evolution” claim.
- More importantly, Plan2Evolve in practice requires strong teacher models to create the symbolic domains and reasoning traces. This dependency undermines the claim of self-evolving (it is not "self-" if it relies on a teacher!).

(2) My main issue is with the framework is the need for the LLMs to produce the domains by themselves. In practice, the success of Plan2Evolve depends heavily on the model’s ability to create valid and meaningful problems. It is unclear how far this can be pushed. Also, it is unclear how this process is done. This part was forwarded to the paper by Huang et al. (2025b), but I think it should be contained in the main text of this work.

(3) The experiments give solid results (as I mentioned above) but some of the design choices for the framework would need extra experiments to be better motivated. As explained in (2), an important aspect of the framework is that the LLMs create the domains they will be trained on. But then, in the experiments, some of the domains used already exist. What am I missing here? On a similar note, the paper also mentions using FF to produce the training pairs, an off-the-shelf planner. If we rely on an off-the-shelf tool, how is this self-evolving?

(4) The high-level results of the paper --- using output from a stronger teacher to bootstrap reasoning traces helps reasoning ---   is known (e.g., see [1] below). The cool thing of this submission would be the self-evolving aspect of it but, as argued in the previous points, the claim of really being self-evolving is too weak.

(5) The paper is too hand-wavy for an ICLR paper. For example

- Section 3.1 is very sloppy. Predicates are described as Boolean classifiers (they are relations), actions lack explicit add/delete semantics, and goals are specified as complete states (which is quite limiting).
- Section 3.3 is very unclear and high-level; from the textual description it is not obvious what is going on
- The paper does not report any of the hyperparameters used during supervised fine-tuning. Neither Section 5.4 nor the Appendix provides training details such as optimizer, batch size, learning rate, training steps, or fine-tuning method.
- In general, the paper would be helped by using some of the general guidelines proposed by [2]


[1] Muennighoff et al. "s1: Simple test-time scaling"\
[2] Katz et al. "Make Planning Research Rigorous Again!"

**Questions:**

- Can you clarify what exactly you mean by self-evolving? Could you please explain me why Plan2Evolve is indeed self-evolving despite the criticism that I raised above?

- Have you actually run multiple Plan2Evolve iterations? If not, can you provide evidence or analysis that an additional iteration would further improve results, to justify the self-evolving claim?

---

> ### Author Response · Authors · 2025-12-02
> **Reply to W1-3**
>
> ## W1
>
> We thank you for the time you invested in reviewing our submission and for acknowledging the strengths of our method. We also appreciate your positive assessment of the solid empirical results demonstrated in our experiments. However, we would like to respectfully point out that the review contains a few factual inaccuracies and references to issues that are not present in the paper.
>
> ### What makes it self-evolve:
>
> **No external teacher model is used.**
> Plan2Evolve does not rely on any teacher model. The same backbone LLM both generates the PDDL planning domain and uses the domain to produce the training data that is then used to fine-tune itself. This self-bootstrapping loop does not involve any external supervision or a stronger model. Our key contribution is that we are the first to generate a robotic PDDL planning domain and then use this domain to produce training data, rather than relying on the LLM to generate training data directly. The teacher–student study that appears in the paper is only an ablation study included for completeness, to show that Plan2Evolve can also operate in a teacher–student setting when data is provided by another LLM. It is not part of our main pipeline, and it is not used in any of the core experiments.
>
> **LLM self-generation of PDDL planning domain and dataset:**
> PDDL planning domains encode physical constraints, logical dependencies, object properties, and symbolic actions, enabling automatic generation of physically feasible and logically consistent task plans that serve as high-quality training data [1,2]. Prior work has extensively explored using LLMs for planning domain construction (see main paper L119–122). The generated planning domain thus produces a task-planning dataset to back SFT the LLM itself.
>
> **Why we frame it as self-evolution:**
> As described above, we use the term evolution to describe the process in which the LLM generates data to train itself without external teacher supervision.
>
> ---
>
> ## W2
> We did not include full technical details on domain generation because it is not the primary focus of this work. The feasibility and reliability of using LLMs to construct planning domains on various problems have already been extensively studied in prior research, which we reviewed in the related work section (main paper L119–122). These works collectively demonstrate how LLMs generate planning domains, diagnose and correct domain errors, and improve domain quality through environment-based feedback. Our final domain-generation pipeline builds directly on these established methods and integrates the techniques proposed in [3–5].
>
> The high-level process is:
>
> 1. The LLM generates an initial planning domain using human-demonstrated task-planning processes and task descriptions [3].
> 2. The LLM evaluates the functionality of this domain and corrects any detected errors [4].
> 3. The resulting planning domain is executed to prune unnecessary components and recover important missing elements [5].
>
> These methods are empirically stable across various tasks, and we integrate them into our planning-domain generation module.
>
> ---
>
> ## W3
>
> **All planning domains in our paper are generated entirely by the LLM itself.**
> All planning domains, regardless of whether similar ones exist in prior research, are generated independently by the LLM. The “seen” tasks are those on which the LLM has run the full pipeline (domain generation → data generation → SFT).
> The “unseen” tasks are evaluated directly without running Plan2Evolve on them.
> Thus, “seen vs. unseen” refers to whether *Plan2Evolve* was applied, not whether we used human-made domains.
>
> **Using Search Algorithm does not contradict the self-evolving nature of the framework**
> FF does **not** provide knowledge or reasoning traces. It only performs search over the LLM-generated domain, like a simulator or compiler.
> The knowledge the model learns still originates from itself.
>
> ---

---

> ### Author Response · Authors · 2025-12-02
> **Reply to W4-5 and References**
>
> ## W4
>
> **There is no external stronger model used as teacher in Plan2Evolve.**
> The teacher–student study is **only an ablation** to demonstrate extensibility.
>
> **Differences from s1:**
> 1. s1 enhances performance via **manually curated data** (stated in their abstract). While Plan2evolve automatically sefl-generate the training data.
> 2. s1 relies on **external stronger models** to generate reasoning traces, while Plan2Evolve produce the training traces by the base model itself.
> 3. The two methods are fundamentally different as s1 is a test-time token cost management method while plan2evolve focus on enhancing long-horizontal planning.
>
> ---
>
> ## W5
>
> **Predicate and action definition:**
> We mean to say predicates are characterized by Boolean classifiers as this is commonly seen in many papers[5-8]. To follow a stricter definition formality, we would like to revise it as: object property and relations characterized by a classifier. We acknowledge that a formal logical action definition includes add/delete, we did not include the full definition in the main paper because of the page limit, which is also commonly seen in the paper of related fields[2,9]. We would like to revise the action definition mentioned in the revised version.
>
> **We could not find where we wrote ‘complete states’.**
> We searched and found no such phrasing; please specify lines if any.
>
> **Training details:**
> Appendix B.3 contains **batch size**, **learning rate**, **training steps**, **fine-tuning method**, **prompt template**, and more.
>
> ### References
>
> [1] McDermott et al. Pddl-the planning domain definition language, 1998.
>
> [2] Dalal et al. Imitating Task and Motion Planning with Visual Transformers, 2023
>
> [3] Huang et al., One demo is all it takes: Planning domain derivation with LLMs from a single demonstration, 2025
>
> [4] Guan et al., Leveraging pretrained large language models to construct and utilize world models for model-based task planning, 2023
>
> [5] Silver et al., “Predicate Invention for Bilevel Planning,”, 2023.
>
> [6] Kumar et al “Learning Efficient Abstract Planning Models that Choose What to Predict,”, 2023.
>
> [7] Liang et al., “VisualPredicator: Learning abstract world models with Neuro-Symbolic Predicates for robot planning,” 2024.
>
> [8] Athalye et al, “Predicate invention from pixels via pretrained vision-language models,” 2024.
>
> [9] Lee et al., "STAMP: Differentiable Task and Motion Planning via Stein Variational Gradient Descent", 2025

---

### Meta-Review · Area_Chair_fFLw · 2026-01-05

**Summary:**

This paper presents Plan2Evolve, a self-evolving framework designed to enhance the task planning capabilities of Large Language Models (LLMs) by generating and learning from Chain-of-Thought (CoT) data. Specifically, an LLM is employed to generate symbolic representations in the form of Planning Domain Definition Language (PDDL) from natural language task descriptions. The resulting planning domain and task solution obtained from a PDDL solver are then translated back into natural language with chains of thought. These translated examples collectively form a dataset used for supervised fine-tuning to strengthen the planning capability of the LLM.

**Reviewer Concerns:**

The Reviewer 3hhe concerns about the “self-evolving”  claim,  the dependence of the teacher/solver to create the symbolic domains and reasoning traces, the success of Plan2Evolve depends heavily on the model’s ability, etc.

The Reviewer NQ8s  concern is focused on the following:

(1) **Unconvincing comparison without related baselines:**  the authors select only unfine-tuned LLMs as baseline methods and do not compare their approach with other LLM-based methods.

(2) **Unconvincing justification with only a single backbone LLM:** The proposed method is implemented only on Qwen and not transferred to other base LLMs such as Llama or GPT-4o-mini. I suspect that some LLMs may lack the ability to generate executable planning domains solely from task descriptions.

(3) **The limiting  lengths of planning steps:** he paper does not examine its planning capability on longer-horizon tasks, which limits understanding of its scalability and robustness.

The Reviewer 8Nk8 is also concerned about the claim that the "Self-Evolving" nature depends heavily on the solver's ability, and the limitation of the PDDL framework in its expressiveness, which cannot encompass the full spectrum of real-world planning tasks.

The authors provide additional experiments on Llama-3.1-8B model, but fail to provide a comparison with related baselines as Reviewer NQ8s  required.  In addition, the common concern about the limitation and dependency of the solver's ability is not well addressed.

**Reviewer Scores:**

The Reviewer 3hhe and the reviewer NQ8s gave scores below the acceptance threshold and may not change their scores because some of their concerns have not been well addressed.

---

### Decision · Program_Chairs · 2026-01-26

Reject